# Mixed Layer Height Retrievals Using MicroPulse Differential Absorption Lidar

Luke Colberg <sup>1</sup>, Kevin S. Repasky <sup>1</sup>, Matthew Hayman <sup>2</sup>, Robert A. Stillwell <sup>2</sup>, and Scott M. Spuler <sup>2</sup>

<sup>1</sup>Montana State University, Department of Electrical and Computer Engineering, Bozeman, MT, USA

<sup>2</sup>NSF National Center for Atmospheric Research, Earth Observing Lab, Boulder, CO, USA

**Correspondence:** Kevin S. Repasky (repasky@montana.edu)

Abstract. Accurate measurement of the mixed layer height (MLH) is a key observational capability necessary for many studies in weather forecasting, air quality assessment, and surface-atmosphere exchange. However, continuous MLH monitoring with backscatter lidars remains challenging under complex atmospheric conditions, including cloudy conditions and in the presence of residual layers. This study evaluates two complementary MLH retrieval algorithms using a single MicroPulse differential absorption lidar (MPD): an aerosol-based approach that analyzes aerosol backscatter gradients with a wavelet technique and a thermodynamic technique based on the vertical structure of virtual potential temperature profiles. Both techniques were compared against MLH estimates from radiosondes, a Doppler wind lidar, and a high-resolution weather model using data from the M²HATS field campaign in Tonopah, NV, U.S.A., supplemented by a smaller dataset from Boulder, CO, U.S.A. The aerosol method achieved high temporal resolution and agreement with radiosonde MLH estimates under convective conditions ( $R^2 = 0.819-0.919$ ), but its MLH estimates deviated from other methods during morning and evening transitions due to residual layer interference. The thermodynamic method avoided these problems but had coarser resolution and degraded instrument performance beneath clouds ( $R^2 = 0.661-0.845$ ). Because lidar generally cannot penetrate clouds, conditions with clouds at or below the MLH are not considered, while those with clouds above the MLH are retained. The study highlights the strengths and weaknesses of each method. Together, they offer a path toward more reliable automatic MLH monitoring with a single instrument by capturing when different MLH definitions converge.

#### 1 Introduction

The planetary boundary layer (PBL) is the lowest part of the atmosphere and represents the interface between the land surface and the free troposphere. The PBL plays a vital role in land-atmosphere interactions, like regulating the exchange of energy, moisture, and pollutants (Santanello et al., 2018; Wulfmeyer et al., 2018). By definition, the PBL is the part of the troposphere directly influenced by surface forcings on timescales of about an hour or less (Stull, 1988). PBL meteorology significantly impacts many societal and economic sectors (Teixeira et al., 2021). As such, accurate knowledge of the state and evolution of the PBL is necessary for studying turbulent mixing, convective transport, and other important atmospheric processes (Stull, 1988).

The planetary boundary layer height (PBLH), the vertical extent of the PBL, is a fundamental parameter describing its structure. The PBLH is an important scaling parameter in mixed layer similarity, particularly for turbulence statistics and fluxes in the convective boundary layer (Deardorff, 1972; Kaimal et al., 1976; Deardorff et al., 1980). It serves as a diagnostic tool in evaluating the performance of PBL parameterizations in numerical models (e.g., Coniglio et al., 2013; LeMone et al., 2013; Cohen et al., 2015) and influences turbulence calculations and eddy diffusivities in nonlocal and hybrid closure schemes (e.g., Troen and Mahrt, 1986; Holtslag and Boville, 1993; Hong et al., 2006; Pleim, 2007). The PBLH is particularly important for air quality forecasting because it imposes a vertical limit to mixing and thus strongly affects the dispersion of surface-based pollutants (Banks et al., 2016; Tang et al., 2016; Su et al., 2018). The PBLH is, therefore, a valuable parameter in many studies involving PBL meteorology or air quality.

Unfortunately, the PBLH cannot be directly measured and typically is inferred from identifiable features in the vertical structure of atmospheric variables or passive tracers. Many techniques exist for diagnosing the PBLH (Seibert et al., 2000; Kotthaus et al., 2023, and references therein). Among instruments capable of measuring properties of the PBL, elastic backscatter lidar stands out for its ability to monitor aerosol structure, which acts as a passive tracer for the PBLH, with high temporal resolution across the full diurnal cycle. However, quantitative comparisons between lidar PBLH observations and other methods (e.g., radiosondes or models) can be challenging because retrievals depend on instrumentation, diagnostic techniques, and definitions and assumptions used. As Chu et al. (2019) demonstrated, inconsistencies in PBLH definitions contribute significantly to differences between observations and model outputs. Despite these difficulties, studies by Tangborn et al. (2021) and Dang et al. (2022) demonstrate that assimilated PBLH values can improve forecasts even when the observations are derived from vertical profiles of aerosol backscatter, which is not a model state variable.

The central problem addressed in this study is the lack of continuous, trustworthy PBLH estimates. While backscatter lidars offer continuous coverage, interpreting the vertical structure of aerosols to retrieve physically meaningful estimates of the PBLH remains challenging. A wide range of aerosol backscatter-based retrieval algorithms have been developed to address this challenge (Dang et al., 2019a), and recent work has applied machine learning technologies to improve performance under complex atmospheric conditions (e.g., Rieutord et al., 2021; Liu et al., 2022; Wijnands et al., 2024; Christopoulos et al., 2025). Other recent research has pursued multi-instrument approaches, including synergistic combinations of instruments and intercomparisons of PBLH estimates from different instruments, to provide a more complete picture of PBL dynamics (e.g., de Arruda Moreira et al., 2018, 2019, 2020; Duncan Jr. et al., 2022; Tsikoudi et al., 2022; Smith and Carlin, 2024; Zhang et al., 2025). The sheer number of different algorithms and their refinements highlights the inherent challenge of achieving universally robust retrievals. All of these methods, however, rely on the assumption that aerosols are valid tracers of the PBLH, an assumption that often fails in the presence of multilayer aerosol structure or weak turbulence, and advanced techniques cannot overcome this fundamental limitation.

This paper presents and evaluates two complementary methods for retrieving the mixed layer height (MLH), which generally corresponds to the PBLH in convective conditions, though the presence of a residual layer, the remnant of the previous day's mixed layer, can complicate the relationship and make the exact MLH dependent on definition. Both methods are applied to data from a single MicroPulse DIAL (MPD; DIAL: differential absorption lidar) system, a vertically pointing lidar that

provides profiles of calibrated aerosol backscatter, absolute humidity, and temperature. One method applies a Haar wavelet transformation (HWT) to the aerosol backscatter coefficient profiles, combined with a layer tracking algorithm and a novel top-limiter constrained by the High-Resolution Rapid Refresh (HRRR) (Dowell et al., 2022) model MLH estimate. The other applies a parcel method with a 1 K offset to virtual potential temperature profiles. To our knowledge, this is the first application of the parcel method to lidar data.

The thermodynamic method aligns more directly with the physical definition of the MLH in most conceptual models, while the aerosol method can track rapid changes in boundary layer structure with greater precision and temporal resolution, provided that aerosol stratification marks the MLH. By comparing these techniques against multiple reference datasets and identifying their strengths and limitations, this study aims to improve the usefulness of lidar-derived MLH observations by clarifying the conditions under which each method is most reliable and by supporting future algorithm development. Two case studies highlight the strengths and limitations of each method. Comparisons with radiosonde MLH retrievals assess correlations and biases, and time-series comparisons to the HRRR model and DWL retrievals identify the times of day when each technique performs best.

This paper is organized as follows: Section 2 provides a conceptual model of the PBL, Section 3 describes the datasets and instruments; Section 4 describes the MLH retrieval methods; Section 5 presents and interprets the results, including case studies, the radiosonde comparisons, and time-series analysis; Section 6 discusses performance trade-offs, limitations, and considerations for applying each method; and Section 7 provides concluding remarks.

## 2 Conceptual Model of the PBL

This study adopts a conceptual model of the PBL consisting of several sub-layers, illustrated in Figure 1. Although real-world PBL structures often deviate from this model, it provides a useful framework for interpreting PBL development.

At sunrise, solar radiation heats the surface, generating a positive surface buoyancy flux (i.e., transferring sensible and latent heat from the surface to the atmosphere), causing turbulent mixing and forming a near-adiabatic mixed layer. This layer deepens through the morning and reaches full depth in the mid-afternoon. The mixed layer, also called the convective boundary layer, decays in the evening when the surface buoyancy flux becomes negative. A stable boundary layer typically forms near the surface due to longwave radiative cooling, greenhouse gas absorption, and mechanical shear, while the previous day's mixed layer remains lofted, forming a decoupled residual layer (Lothon et al., 2014). This residual layer is decoupled from the surface and lacks strong convection (though cases of weak mixing have been observed; Fochesatto et al., 2001), yet retains the history of mixing from the previous day. It persists above the stable boundary layer until both are entrained into the growing mixed layer the following day. Its presence and properties can strongly influence mixed layer development during the morning growth period (Blay-Carreras et al., 2014). Note that some researchers include the residual layer in their definition of the PBL, defining the PBLH as its top (e.g., Chu et al., 2022), in contrast to the definition used here. Some researchers (e.g., Collaud Coen et al., 2014) further divide the PBL into regimes such as cloudy convective or neutral boundary layers, but this study does not make those distinctions.

**Figure 1.** Schematic of the typical diurnal cycle of the PBL over land on a clear-sky, convective day (adapted from Stull (1988)). Vertical dashed lines indicate sunrise and sunset times, and the dot-dashed line represents the ambiguity in defining PBL structure during the evening transition. Height is above ground level (AGL).

An interfacial layer called the transition layer exists at the top of the mixed layer. This transition layer is marked by a virtual potential temperature inversion and is usually co-located with an aerosol backscatter gradient formed as cleaner, more buoyant air from above mixes with less buoyant, aerosol-dense air from below. This gradient may be absent if an overlying residual layer has aerosol optical properties similar to the growing mixed layer. The transition layer exists within the broader entrainment zone, defined as the region with negative buoyancy flux, indicating the downward mixing of buoyant air. However, the buoyancy flux cannot be measured with the MPD or with lidars of comparable vertical and temporal resolution, and the entrainment zone spans a broader region than the sharp gradients typically observed by lidar (Brooks and Fowler, 2012). For this reason, this study avoids using the term "entrainment zone" in the context of lidar observations and instead uses "transition layer," which can be observed by lidar instruments in favorable conditions. The capping inversion, located at the top of the residual layer, is marked by a strong virtual potential temperature inversion and a sharp aerosol gradient. While the transition layer is characterized by active entrainment and convective mixing, the capping inversion is more stable, with limited mixing driven by mechanical shear and gravity waves.



The vertical extent of the mixed layer is called the MLH. The term PBLH refers more broadly to the full PBL depth, encompassing both the MLH in convective conditions and the stable boundary layer height in stable conditions (Stull, 1988). The mixing height is a related term primarily used for air quality forecasting, which refers to the height to which surface-based pollutants disperse, effectively representing the vertical extent of active turbulence (Seibert et al., 2000; Tucker et al., 2009). While it is often assumed to be equivalent to the MLH in convective conditions, turbulence does not always extend to the

transition layer, leading to differences (Grimsdell and Angevine, 2002; Träumner et al., 2011; Schween et al., 2014). It is also defined in stable conditions when no mixed layer is present and usually differs from the stable boundary layer height (Tucker et al., 2009; Bonin et al., 2018). Despite the differences in definition, the same diagnostic methods are often appropriate for both the MLH and the mixing height in convective conditions. For example, wavelet transformations of the aerosol backscatter profile have been used in studies retrieving both the MLH (Cohn and Angevine, 2000; Brooks, 2003) and the mixing height (Haeffelin et al., 2012; Schween et al., 2014). Many of the references (e.g., Seibert et al., 2000) discuss methods for diagnosing the mixing height. In this paper, the term MLH will be used for the lidar PBLH observations to distinguish them from the stable boundary layer height, typically below the minimum range of the lidars used in this study. A glossary of terms is shown in Table 1.

Table 1. Glossary of key terms.



| Term                         | Definition                                                                                                          |
|------------------------------|---------------------------------------------------------------------------------------------------------------------|
| PBL                          | Lowest part of the troposphere directly influenced by surface forcing.                                              |
| Mixed layer                  | Turbulent, well-mixed layer typically present during daytime convection; also called the convective boundary layer. |
| Stable boundary layer        | Stratified boundary layer under stable conditions, often at night.                                                  |
| Residual layer               | Remnants of the previous day's mixed layer that remains lofted after convective mixing stops                        |
| Free troposphere             | Region above the PBL and residual layers, minimally influenced by the surface.                                      |
| Transition layer             | Interface between the mixed layer and the free troposphere or residual layer, typically marked                      |
|                              | by a virtual potential temperature inversion and an aerosol gradient.                                               |
| Entrainment zone             | Region near the top of the mixed layer with negative buoyancy flux, indicating the downward                         |
|                              | mixing of buoyant air; encompasses the transition layer.                                                            |
| Capping inversion            | Sharp virtual potential temperature inversion and aerosol gradient that caps the residual layer                     |
|                              | and separates it from the free troposphere.                                                                         |
| PBLH                         | Vertical extent of the PBL, including both the MLH and the stable boundary layer height.                            |
| MLH                          | Vertical extent of the mixed layer; also called the convective boundary layer height.                               |
| Stable boundary layer height | Vertical extent of the stable boundary layer.                                                                       |
| Mixing height                | Height to which surface-based pollutants disperse, representing the vertical extent of active                       |
|                              | turbulence. Commonly used in air quality forecasting. Often equivalent to the MLH in convec-                        |
|                              | tive conditions, but also defined under stable conditions when no mixed layer is present.                           |

An underlying assumption of aerosol-based MLH retrievals is that aerosols act as passive tracers of boundary layer dynamics. We adopt the convention that aerosols are broadly classifiable by size: "nucleation" mode particles ( $< 0.1 \,\mu\mathrm{m}$ ), "accumulation" mode particles ( $0.1-1 \,\mu\mathrm{m}$ ), and "coarse" mode particles ( $> 1 \,\mu\mathrm{m}$ ). Typically, nucleation mode particles are short-lived because they rapidly coagulate into larger particles, and coarse mode particles are likewise short-lived due to efficient gravitational settling. In contrast, accumulation mode particles have a relatively small settling velocity and low coagulation rates, allowing

them to persist in the atmosphere for weeks to months, dominating the aerosol population. The relatively small settling velocity of accumulation mode aerosols also means their vertical transport is dominated by turbulent mixing (Pandis et al., 1995).

Daytime turbulent motions rapidly homogenize aerosols within the mixed layer, and at night, their weak gravitational settling allows them to remain suspended in the residual layer in the absence of turbulence. Because mixed layer air generally does not penetrate into the free troposphere (Stull, 1988), aerosol concentrations above the MLH or capping inversion are expected to be low, producing sharp gradients in aerosol concentration. Many aerosol-based MLH retrievals rely on identifying these gradients in aerosol lidar profiles (Dang et al., 2019a).

# 3 Datasets and Instrumentation

#### 3.1 Datasets





The primary dataset for this study was collected during the multi-point Monin–Obukhov similarity horizontal array turbulence study (M²HATS) (Tong et al., 2025), held near Tonopah, Nevada, USA (38.0486°N, 117.0863°W; elevation 1641 m). The lidar was located in the delta of a dry creek draining toward a dry lake bed, with no nearby buildings, where the terrain varied by less than 10 m in elevation within 2 km of the site. Foothills were located about 10 km away, and mountains with elevations exceeding the experimental site elevation by over 1000 m were approximately 35 km away. Vegetation around the site consisted of sparse grass and sagebrush. The subset of data used in this paper spans from 23 July to 12 September 2023. During summer, Tonopah's climate is characterized by low humidity, sparse cloud cover, and intense solar radiation. The average maximum daily MLH values for June–August in this region, derived from CALIPSO (McGrath-Spangler and Denning, 2012) and GNSS (Global Navigation Satellite System) radio occultation (Kalmus et al., 2022), exceed 2.5 km AGL. In Tonopah, aerosols consist of a mix of particulate organic matter, fine dust, and ammonium sulfate (in decreasing order of contribution) with occasional heavy organic and elemental carbon loading from wildfires (Hand et al., 2024).

A secondary dataset was collected in Boulder, Colorado, USA (40.0378°N, 105.2415°W; elevation 1615 m), over a four-day period from 9 to 12 March 2025. Conditions during this period were cooler than the arid summer of Tonopah. In Boulder, springtime aerosols include a mix of particulate organic matter, fine dust, and ammonium sulfate (in decreasing order of contribution) (Hand et al., 2024), together with urban aerosols from traffic and industrial emissions (Boucher, 2015), contributed locally by Boulder and the nearby Denver metropolitan area. Boulder lies about 5 km east of the Front Range, with open plains to the east. The lidar was set on the edge of the city near a small lake, with surrounding vegetation consisting of lawns and trees.

In both locations, the presence of nearby mountain ranges can cause local mountain-plains interactions and lee waves generated by advection over the ranges. While these effects may influence mixed layer development, they were not isolated in this study. The short dataset from Boulder was included to verify that the retrieval algorithms presented are not overfit to the conditions seen during M<sup>2</sup>HATS.

## 3.2 The MPD

Lidar measurements of the aerosol backscatter coefficient and virtual potential temperature were made using an NCAR (National Center for Atmospheric Research) MPD (NCAR, 2023). The MPD employs a combination of DIAL and high spectral resolution lidar (HSRL) techniques. The DIAL techniques provide vertical profiles of absolute humidity and temperature. Virtual potential temperature is calculated from the temperature and humidity profiles. The HSRL technique measures vertical profiles of calibrated aerosol backscatter at a particular wavelength (770 nm for the MPD). In contrast, ceilometers and elas-160 tic backscatter lidars measure attenuated backscatter. An inversion technique (e.g., Fernald et al., 1972; Klett, 1981), which requires overlap correction, relies on assumptions about aerosol properties, and contains coupled backscatter and extinction terms, is required to isolate the aerosol backscatter, introducing further uncertainty. The HSRL relies only on an internal calibration of the spectral response of the two receiver channels and provides calibrated aerosol backscatter directly. Most HSRLs require a differential overlap correction between the combined and molecular channels, but the MPD measures this correction directly as part of its retrieval process (Stillwell et al., 2020; Hayman et al., 2024). The reduced reliance on assumptions and external calibrations makes the MPD HSRL more robust for high-quality MLH retrievals. Further details about the instrument hardware can be found in Spuler et al. (2015, 2021), Hayman and Spuler (2017), and Stillwell et al. (2020), with data processing following Hayman et al. (2024). The instrument operated continuously throughout the collection period. In the configuration deployed, the MPD had a minimum usable range of 318 m AGL (set by quality control; limited by detector recovery from the outgoing pulse; see Stillwell et al., 2025), an effective vertical resolution of 150 m (primarily set by the pulse length, though retrieval dependent; see Hayman et al., 2024), and a range bin spacing of 37.5 m from the 250 ns digitization, which oversamples the pulse. The maximum range is scene-dependent, typically about 6 km AGL or to cloud base. Aerosol backscatter, water vapor, and temperature profiles were produced at five-minute intervals, with effective temporal resolutions of 5 minutes for aerosol backscatter, 10 minutes for water vapor, and 40 minutes for temperature, as determined by the retrieval processing 175 described in Hayman et al. (2024). The MPD also includes a built-in weather station (Lufft WS300), which provides the surface observations required for the MPD-thermodynamic method.

## 3.3 Other Instrumentation



The M<sup>2</sup>HATS dataset was collected alongside a suite of other instruments, including a DWL, radiosondes, a surface weather station, precipitation sensors, and a flux tower array (NCAR, 2023). The DWL (Halo-Photonics Streamline XR) (Pearson et al., 2009) operated in vertical staring mode throughout the data collection period, and the data was processed using the manufacturer's proprietary software. For this study, DWL data were integrated into ten-second profiles with 30 m range bins. Radiosondes (Vaisala MW41 / RS41) were launched daily at approximately 17:00 and 22:00 UTC (10:00 and 15:00 local time), except during Tropical Storm Hilary (NHC, 2023). Surface conditions for the radiosonde MLH retrievals were provided by the Integrated Sounding System (ISS) smart weather sensor (Lufft WS800-UMB). Precipitation data from this sensor and a disdrometer (OTT Parsivel<sup>2</sup>) were used to identify periods of precipitation so they could be removed from the DWL retrievals, since scattering from raindrops increases the vertical velocity variance and prevents reliable MLH retrievals. Both were part of

the ISS. Additionally, surface and flux measurements from the Integrated Surface Flux System (ISFS) tower array (Tong et al., 2025) were used to calculate buoyancy fluxes for the DWL MLH retrieval.

The Boulder dataset consisted solely of MPD measurements, with no additional instrumentation for reference MLH esti-190 mates.

#### 4 Methods





The MLH was estimated in this work using retrieval methods applied to three observing systems (MPD, DWL, and radiosondes) and the HRRR model. Each method diagnoses the MLH using observable aerosol, thermodynamic, or turbulence profiles. Assumptions linking these observations to the conceptual model in Section 2 are highlighted. When possible, methods were implemented consistently across platforms to enable direct comparisons. The following subsections describe each retrieval method.

## 4.1 MPD-Aerosol Method

The MPD-aerosol method, assumes that aerosols are a good passive tracer of the mixed layer due to turbulent mixing. The method tracks aerosol gradients with constraints to limit the false detection of lofted residual layers. Colberg et al. (2022) demonstrated MLH retrievals using an HWT method and data from a similar lidar instrument; the method employed here builds on this work. The key differences used here and that of Colberg et al. (2022), a layer tracking algorithm and a different top limiter, are discussed in detail below.

The algorithm begins by detecting and removing cloudy data as described in Colberg et al. (2022). Next, the HWT (Gamage and Hagelberg, 1993) is applied to the aerosol backscatter coefficient, as described in Appendix A. The HWT can be understood as a gradient operator with a low-pass filter (Comerón et al., 2013). The HWT detects regions of steep change and has been widely used to identify the MLH in aerosol backscatter profiles (e.g., Davis et al., 2000; Cohn and Angevine, 2000; Brooks, 2003; Baars et al., 2008). The challenge in diagnosing the MLH is selecting the correct layer. After the HWT, Dijkstra's algorithm (Dijkstra, 1959) is applied to track layers over time using the Pathfinder method described by de Bruine et al. (2017). Based on the stated conceptual model, the MLH is assumed to be continuous. This Pathfinder algorithm converts the problem of tracking the MLH into an optimization problem that identifies the most likely continuous path in time. It is initialized using the strongest HWT peaks to identify the tops of all aerosol layers. For this study, the Pathfinder method was independently implemented and applied to the HWT of calibrated aerosol backscatter profiles averaged over five minutes. Implementation details are provided in the accompanying code (see Code Availability section).

A top limiter, similar to methods of Gan et al. (2011) and Dang et al. (2019b), is defined based on the time of day and the
HRRR model MLH estimate. From sunrise to midday (halfway between sunrise and sunset), if the HRRR MLH is less than
1000 m AGL, the top limiter is the HRRR MLH plus 500 m; if greater than 1000 m, the top limiter is set to one and a half
times the HRRR MLH. From midday to sunset, it is fixed at 6000 m. This limiter only bounds the search region. It helps to
avoid early morning selection of lofted residual layers instead of a shallow mixed layer and is effectively disabled after midday

since 6 km exceeds the deepest MLH observed in the dataset. The limiter guides the search toward the correct MLH by using the HRRR MLH as a flexible, adaptive reference rather than imposing a rigid cutoff. Layers above the top limiter are discarded, and the peaks in the HWT profile nearest to the remaining layers are then used to reinitialize the Pathfinder algorithm. Then, the lowest significant layer in each profile is located, and the closest peak in the HWT is diagnosed as the MLH. MLH estimates above low clouds or near precipitation were removed, since retrievals above these features are unreliable. MLH estimates below cloud bases were retained.

Decoupled low-altitude aerosol layers can still cause false MLH detections. Additionally, the algorithm is not run at night, as the only observable aerosol layers are assumed to be residual layers. Accordingly, this method is only applied between local sunrise and local sunset. Prior studies (e.g., Sawyer and Li, 2013; Bonin et al., 2018) have shown that aerosol stratification in stable nocturnal boundary layers is typically decoupled from thermodynamic and turbulence-based indicators of the boundary layer height. Finally, the minimum detection range of the MPD-aerosol method is 412 m, a combination of the minimum observable altitude, the range resolution of the HSRL observations, and the width of Haar wavelets.

# 4.2 MPD-Thermodynamic Method







The MPD-thermodynamic method diagnoses the MLH using a parcel method (Holzworth, 1964) applied to virtual potential temperature profiles. This method assumes a convectively unstable atmosphere driven by surface heating, in which the mixed layer has near-constant virtual potential temperature (i.e., buoyancy), and the transition layer contains an inversion with a sharp increase. The MLH is diagnosed as the altitude where a rising parcel from the surface becomes neutrally buoyant, where its virtual potential temperature equals the surface value. This method is widely used with radiosondes. It requires accurate surface observations, which are provided by the MPD's built-in weather station. The MPD data are self-contained and do not import data from radiosondes or weather models.

Applying the parcel method to the MPD profiles presents challenges due to instrument noise. A simple implementation that selects the first altitude where the virtual potential temperature exceeds the surface value is prone to false detections when applied to the MPD data, and high-altitude noise can lead to unrealistically deep MLH estimates. To mitigate these issues, empirical thresholds and a top-down search strategy were used. Full details of the methodology are included in Appendix B. In brief, the MLH is diagnosed with a 1 K offset relative to the surface virtual potential temperature bounded by a 3 K top limiter. MLH estimates retrieved above clouds or precipitation were excluded as the temperature retrieval becomes unreliable in these conditions, while MLH estimates below cloud bases were retained.

The parcel method has known limitations in neutral and stable conditions (Hennemuth and Lammert, 2006) where its underlying assumptions are not valid.

# 4.3 Radiosonde MLH Methods

Radiosonde profiles provide many different methods for determining the PBLH, though different diagnostic methods can lead to differences of several hundred meters (Seidel et al., 2010). This study used two methods: the bulk Richardson method (Seidel et al., 2012; Guo et al., 2021) and the parcel method with a 1 K offset, applied identically to the MPD-thermodynamic

method, except the surface virtual potential temperature was taken from the first valid in-air sonde level (typically about 10 m AGL), since surface observations at launch are often unreliable. Radiosondes do not measure aerosol backscatter, so a direct comparison with the MPD-aerosol method is impossible. The bulk Richardson method, identified by Seibert et al. (2000) as the most appropriate method for radiosonde profiles with wind and temperature measurements, is used to diagnose the MLH for comparison with the MPD-aerosol method. Details of the bulk Richardson number calculation, including the formulation of virtual potential temperature and the critical thresholds, are provided in Appendix C. In brief, the MLH is diagnosed as the lowest altitude where the bulk Richardson number exceeds a critical value of 0.25 (Seibert et al., 2000; Seidel et al., 2012; Guo et al., 2021).

It is important to note that radiosonde retrievals are not ground truth measurements. Radiosonde MLH retrievals can misidentify the tops of residual layers as the MLH, and comparisons with lidar data are complicated by horizontal drift as the balloon ascends (the conceptual model assumes horizontally uniform conditions). However, three-dimensional variability may cause the sonde to sample conditions that differ from those directly above the lidar. For example, a radiosonde could ascend through an updraft that raises the observed MLH while the lidar samples an adjacent downdraft that lowers it. In convective conditions, this can produce differences of several hundred meters. Schween et al. (2014) noted that radiosondes tend to be drawn into converging flows (i.e., updrafts) and away from regions of divergence (i.e., downdrafts), potentially introducing a bias toward sampling updrafts, which would raise the MLH retrievals. However, the magnitude of this bias is unclear, and a more rigorous study is needed to assess its implications for using radiosonde profiles as a validation source.

#### 4.4 DWL MLH Method




Vertical velocity variance profiles measured by DWLs can be used to diagnose the MLH (Tucker et al., 2009; Barlow et al., 2011; Schween et al., 2014). The vertical velocity variance is extracted from the vertical wind data using the technique described by Lenschow et al. (2000), and the MLH is diagnosed as the first altitude where the variance falls below a specified threshold. Uncertainty in the vertical velocity variance was estimated following Lenschow et al. (2000). Periods containing data within the mixed layer where the estimated uncertainty was comparable to or exceeded the vertical velocity variance were flagged and excluded from further analysis. To avoid bias, the approach erred on the side of discarding marginal data. It is more precise to say that the DWL tracks the mixing height, which corresponds to the vertical extent of active turbulence. However, this quantity is commonly used as a proxy for the MLH. The retrievals used in this study follow previous approaches with one key difference: the threshold is based on a mixed layer similarity relationship using a convective velocity scale calculated directly from observed surface buoyancy fluxes measured by the ISFS tower array. This approach allows the threshold to respond to actual surface forcing. Details of this method are provided in Appendix D.

Precipitation interferes with DWL retrievals, so periods of rain identified by precipitation sensors were automatically excluded, and data with virga were manually removed. Virga was identified through inspection of the vertical velocity and backscatter fields, where it appears as a region of consistent negative vertical velocity and enhanced backscatter beneath a cloud. The affected periods were excluded from the analysis.

## 285 4.5 HRRR MLH Methods

The HRRR forecast (Dowell et al., 2022), operated by NOAA, provides hourly updated short-term forecasts at 3 km resolution across the continental United States. It uses the Mellor-Yamada-Nakanishi-Niino eddy-diffusivity/mass-flux PBL scheme (Olson et al., 2019) and diagnoses the PBLH, including both the stable PBLH and the MLH, using a hybrid method dominated by the potential temperature profile for stable conditions and the turbulent kinetic energy profile for neutral and unstable conditions. The native PBLH estimate is reported as a continuous variable in meters AGL. Additionally, a second MLH estimate was computed from the HRRR virtual potential temperature profiles, which, in the format used here, are available every 25 hPa in the boundary layer (corresponding to about 250 m vertical spacing at an elevation of 1600 m above sea level, similar to the Tonopah and Boulder sites). These profiles, together with the HRRR surface fields, were used with the 1 K offset parcel method (applied identically to the MPD-thermodynamic method; see Appendix B), enabling direct comparisons.

## 295 5 Results




Observations from the M<sup>2</sup>HATS dataset are analyzed both as detailed case studies and in aggregate. Two case studies are presented: one that illustrates canonical PBL development consistent with the conceptual model and one that does not. Aggregate statistics include comparisons with radiosonde launches and time series data spanning the full data collection period.

#### 5.1 Case Studies

#### 300 5.1.1 6 September 2023

This section presents a case from 6 September 2023, comparing MPD-aerosol and MPD-thermodynamic MLH estimates to those from radiosondes, the DWL, and the HRRR model. Clear skies and weak synoptic forcing allowed mixed layer development beneath a residual layer, conditions consistent with the conceptual model. Figure 2 shows the MLH retrievals from all methods discussed, allowing a direct comparison of their performance throughout the day.

During the early morning, all three lidar-based methods are unable to detect a mixed layer, which is consistent with a stable PBL. Around 8:00, the MPD-thermodynamic and DWL methods detect the onset of mixed layer growth driven by surface heating and turbulence. The MPD-aerosol method initially follows a low-altitude layer near 9:00 that does not rise with morning heating and is higher than the other retrievals during a period when aerosol stratification is less reliable. Due to the high MLH estimates and it not growing in harmony with surface fluxes, this layer is interpreted as a residual layer. Around 10:00, the MPD-aerosol method begins tracking the growing mixed layer.

As the lofted aerosol layer is entrained, the mixed layer grows to 1.9 km by noon. All retrievals capture this growth. During the morning growth period, the MPD-thermodynamic and radiosonde offset parcel methods report higher MLH estimates than the MPD aerosol, DWL, and bulk Richardson methods. This behavior is partly due to the offset parcel method's known upward bias, but additional factors contribute. The 10:00 radiosonde profile shows less than a 1 K increase in virtual potential temperature between 800 m and 1.7 km, indicating a weak inversion at the top of the residual layer. This situation supports

Figure 2. Time series of MLH retrievals for a representative PBL evolution from 6 September 2023. Top: Aerosol backscatter coefficient at 770 nm, overlaid with MLH retrievals from the MPD-aerosol, MPD-thermodynamic (MPD-TD), DWL, radiosondes using the bulk Richardson (BRN) and offset parcel (surface + 1 K) methods, and the HRRR model. Middle: Difference between atmospheric and surface virtual potential temperature overlaid with MPD-thermodynamic and radiosonde offset parcel MLH retrievals. Bottom: Vertical velocity variance overlaid with DWL and radiosonde bulk Richardson method MLH retrievals. Marker density is downsampled for readability and does not reflect the native temporal resolution of each method. Vertical dashed lines indicate sunrise and sunset times. All heights are in AGL.

rapid mixed layer growth until the transition layer merges with the capping inversion. The MPD-thermodynamic method follows this growth until 11:00, when it abruptly shifts to a higher altitude. This jump coincides with a cold bias in the MPD temperature retrieval centered near 1 km, visible as a horizontal band in the center panel of Figure 2. Once this cold bias is exceeded around 11:00, the next cold region is at the capping inversion. Because the conceptual model assumes a continuous MLH, and both the MPD-aerosol and DWL estimates show smooth evolution, this discontinuity is interpreted as an artifact of the 1 K offset rather than a real atmospheric transition. The MLH continues to grow gradually through the afternoon, reaching a maximum height of  $2.4 \, \mathrm{km}$  by 16:30.






Around midday, the convective mixed layer reaches its maximum depth. All retrievals capture this, with the DWL, MPD-aerosol, and 15:00 radiosonde in close agreement. The MPD-thermodynamic method is consistently biased about 250 m higher than the other observations throughout the afternoon. The HRRR model closely follows the observed MLH evolution.

The DWL MLH retrievals decrease during the evening transition while the MPD-aerosol method remains lofted, continuing to track the top of the elevated residual layer. This divergence is well documented (e.g., Grimsdell and Angevine, 2002; Träumner et al., 2011; Schween et al., 2014) and reflects that the DWL measures active mixing, while the MPD aerosol method captures the history of prior mixing. The MPD-thermodynamic MLH estimates also decrease, demonstrating the method's ability to capture the transition. This behavior is consistent with the results of Wang et al. (2012) and Collaud Coen et al. (2014), who demonstrated similar performance using microwave radiometers, and with Summa et al. (2013), who showed comparable results with temperature profiles from a Raman lidar. In all three studies, aerosol lidar retrievals tended to follow the residual layer. The turbulence and thermodynamic indicators are more directly tied to the physical processes that define the mixed layer, whereas aerosols are passive tracers. When persistent disagreement occurs during the evening, the MPD-aerosol method is interpreted to be tracking a residual layer.

This case highlights the strengths and limitations of both MPD retrieval methods. The MPD-aerosol method tracks aerosol layers precisely but can misidentify decoupled stratification and residual layers as the MLH, particularly during morning growth and the evening transition. In contrast, the MPD-thermodynamic method successfully captures the low-altitude morning MLH and the evening transition but is limited by coarser resolution and sensitivity to instrument biases, and the  $1\,\mathrm{K}$  offset causes MLH estimates to be consistently higher than those from other observations.

In this case, the DWL retrievals were consistent with the expected MLH based on the conceptual model of mixed layer evolution and showed no indications of falsely identifying residual layers. However, conditions were ideal for the DWL, with plenty of aerosols and a moderate MLH. Across the full collection period, the DWL signal failed to reach the MLH on 17 of 52 days due to deep PBLs (> 2.5 km) and low aerosol loading, limiting data availability despite strong performance when the signal was sufficient. The maximum range could potentially be extended above the MLH in these cases using advanced post-processing algorithms (e.g., Vakkari et al., 2019), but these methods were not applied here. The MPD methods, on the other hand, have no issues with deep PBLs.

The HRRR model also performed well on this day. However, it can struggle during severe weather. For example, on the day preceding Tropical Storm Hilary, the model's MLH output exceeded other retrievals by nearly 1.5 km in the afternoon. This

estimate was unsupported by any other method and was inconsistent with the observed vertical structure, suggesting that the HRRR prediction did not reflect the actual mixed layer depth.

## 5.1.2 8 September 2023






This section examines a case from 8 September 2023. As in Section 5.1.1, the MPD methods are compared to radiosonde, DWL, and HRRR MLH retrievals. However, this case shows atypical behavior. A large, aerosol-dense plume is advected into the region and entrained into the growing mixed layer with relatively low aerosol loading. Figure 3 illustrates how each retrieval method responds as the plume alters the mixed layer.

During the early morning, a 1.5 km deep residual layer exists over a stable layer. Between 7:30 and 10:00, the MPD-aerosol tracks low-altitude aerosol stratification between 450 m and 800 m. However, the absence of observed mixing by the DWL suggests that these layers are decoupled from the surface. The 10:00 radiosonde shows a shallow mixed layer, with the transition layer extending from approximately 400–900 m and MLH estimates ranging from 393 m (MPD-thermodynamic) to 862 m (radiosonde offset parcel). From 10:00 to 13:30, the methods show reasonable agreement, tracking the growing mixed layer to a height of 1200 m by 13:30.

Beginning at about 11:30 an aerosol-dense plume begins to advect over the site, with its leading edge first visible at 3.2 km. At 13:30, the plume extends from 1.5 km to 3.5 km. Its aerosol optical properties are denser than the mixed layer's, and its virtual potential temperature (i.e., buoyancy) is nearly identical. At 13:30, the lofted plume begins to be entrained into the mixed layer. The DWL tracks continuous growth, from 1.2 km at 13:30 to 2.5 km at 16:25.

The MPD-aerosol and MPD-thermodynamic methods cannot track turbulence and remain near 1.2 km until just before 15:00, when they jump to 3.3 and 3.5 km, respectively. The 15:00 radiosonde retrieves an MLH of 3.7 km using both the bulk Richardson and offset parcel methods, showing reasonable agreement with the MPD retrievals. The DWL retrieved an MLH of 1.9 km. Radiosonde profiles show a weak inversion at 1.9 km, with only a 0.35 K increase in virtual potential temperature and no significant turbulence above. The HRRR estimate lies between the MPD and DWL estimates, though it cannot be expected to capture the dynamics of a localized plume.

Determining which estimate represents the true MLH depends on its definition and this case is especially ambiguous. In this case, the DWL tracks the mixing height rather than the MLH. In many situations, these are similar, but here, they diverge because turbulence does not extend through the full depth of the thermodynamically mixed layer. The DWL estimate of 1.9 km reflects the depth of turbulence, while the air up to 3.7 km has nearly the same buoyancy as at 1.9 km, making the boundary between air masses poorly defined and easily deformed. This causes the rapid growth in MLH from 13:30 to 16:25. If the MLH is defined instead by the altitude of the first increase in atmospheric stability, then 3.7 km is more appropriate since the plume and the turbulent mixed layer have nearly identical buoyancy.

Between 17:00 and 18:15, the MPD-aerosol and DWL methods show close agreement as the mixed layer contracts, and the MPD-thermodynamic method is 500–700 m higher. The same evening MLH decoupling, noted in Section 5.1.1, is also present in Figure 3. The MPD-aerosol method remains at the residual layer. The MPD-thermodynamic method drops below 500 m between 18:30 and 19:00, capturing the transition. The DWL method stops tracking the MLH altogether, as turbulence

Figure 3. Time series of MLH retrievals for an atypical PBL evolution from 8 September 2023. **Top:** Aerosol backscatter coefficient at 770 nm, overlaid with MLH retrievals from the MPD-aerosol, MPD-thermodynamic (MPD-TD), DWL, radiosondes using the bulk Richardson (BRN) and offset parcel (surface + 1 K) methods, and the HRRR model. **Middle:** Difference between atmospheric and surface virtual potential temperature overlaid with MPD-thermodynamic and radiosonde offset parcel MLH retrievals. **Bottom:** Vertical velocity variance overlaid with DWL and radiosonde bulk Richardson method MLH retrievals. Marker density is downsampled for readability and does not reflect the native temporal resolution of each method. Vertical dashed lines indicate sunrise and sunset times. All heights are in AGL.

vanishes simultaneously at all altitudes rather than decaying from the top of the mixed layer downward, as is more commonly observed. Although the DWL requires aerosols for wind speed retrievals, the vertical velocity variance dropped below the MLH threshold before the signal was lost throughout the entire day, indicating that the MLH retrievals were not affected by signal dropout.

This case highlights how MLH estimates diverge when the mixed layer evolution differs from the conceptual model. Two key assumptions of that model are not met: that surface-driven convection is the primary source of aerosols and that a significant inversion coincides with the vertical extent of turbulence. In contrast to the more typical case in Section 5.1.1, where the methods more closely agree, this case demonstrates how advection and entrainment of a lofted plume can decouple thermodynamic and aerosol indicators of the MLH from the vertical extent of turbulence. The MPD-aerosol and MPD-thermodynamic methods do not show continuous growth but instead jump to the top of the plume. This jump reflects the mixing of air masses rather than surface-driven growth and appears nonphysical when viewed in the framework of the conceptual model. The DWL method, in contrast, does not reflect the full vertical extent of the well-mixed atmosphere because turbulence does not reach the top of the mixed layer, and the method assumes that it does. The case demonstrates the limitations of relying on a single MLH estimate and highlights the added value and nuance that can be gained using multiple observational platforms.

# 5.2 Radiosonde Comparisons







This section evaluates the performance of the MPD MLH retrieval methods by comparing them to MLH estimates from radiosondes. The goal is to assess performance under different conditions and identify when retrievals diverge.

The MPD-aerosol retrievals are compared with radiosonde estimates using the bulk Richardson method, and the MPD-thermodynamic retrievals are compared to radiosonde estimates using the offset parcel method, which applies the same diagnostic technique and allows direct comparison. During the M²HATS data collection period, 98 radiosondes were launched, 48 at 17:00 UTC (10:00 local) and 50 at 22:00 UTC (15:00 local). The 17:00 UTC launch typically captured the morning growth phase, often beneath a residual layer, while the 22:00 UTC launch captured the fully developed mixed layer. Both launch times occurred during daytime convective conditions within the valid operating range of the MPD retrieval algorithms (sunrise to sunset). For each radiosonde, the lidar profile and MLH estimate used in the comparison were those closest in time to when the radiosonde ascent passed through the lidar-derived MLH, instead of the launch time, which could introduce a mismatch due to the radiosonde ascent time. These times are favorable for the MPD retrievals, as convective conditions are typically present, and the MLH is usually above the minimum range. Launches where the MPD methods could not retrieve the MLH were excluded, which occurred in 23 cases for the MPD-aerosol method and 14 for the MPD-thermodynamic method. These cases are excluded primarily due to low clouds and precipitation, which the MPD cannot penetrate. In 9 cases, the MPD-thermodynamic method reported an MLH while the MPD-aerosol method did not, identifying a false MLH beneath boundary layer clouds.

Figure 4 shows scatter plots comparing MPD and radiosonde MLH retrievals across increasingly restrictive subsets. Panels (a)–(c) compare the MPD-aerosol to the bulk Richardson method, while panels (d)–(f) compare the MPD-thermodynamic to the radiosonde offset parcel method. Panels (a) and (d) show all comparisons, panels (b) and (e) show clear-sky conditions, and panels (c) and (f) show clear-sky conditions at 15:00 local time. Clear-sky conditions are defined as those with no cloud

detected directly above the lidar during the time the radiosonde ascent passed through the MPD-derived MLH, though nearby clouds outside the lidar beam may have been present.

**Figure 4.** Comparison of MPD-derived MLH retrievals with radiosonde-based retrievals (all values in meters AGL). Panels (a)–(c) show MPD-aerosol retrievals compared with radiosonde bulk Richardson retrievals under different conditions: (a) all conditions, (b) clear-sky conditions, and (c) clear-sky conditions at 22:00 UTC (15:00 local). Panels (d)–(f) show MPD-thermodynamic retrievals compared with radiosonde offset parcel retrievals under (d) all conditions, (e) clear-sky conditions, and (f) clear-sky conditions at 22:00 UTC. The dashed black line indicates 1:1 agreement, and the solid red line shows the linear regression fit.

The MPD-thermodynamic retrievals show reasonable correlations with radiosonde estimates, with  $R^2$  values from 0.661 to 0.845. The bias (MPD-sonde) of -395 m likely results from degraded MPD temperature profiles in cloudy conditions, which cause the parcel threshold to be exceeded beneath clouds at the MLH. In cloudy cases alone (not shown),  $R^2$  drops to 0.497, and the bias increases to -855 m. Restricting the comparison to clear-sky cases reduces the biases. The strongest correlation occurs for all-sonde, clear-sky comparisons ( $R^2 = 0.845$ ; bias = -165 m). Afternoon sondes yield  $R^2 = 0.661$  with a reduced bias of -71 m. A direct comparison of the two radiosonde methods (not shown) yields  $R^2 = 0.963$ , with the offset parcel method averaging 250 m higher than the bulk Richardson method. This difference arises primarily from the 1 K offset applied to the parcel method. The standard parcel method (no offset) corresponds to the height where the bulk Richardson number is


0 (see Appendix C). With the  $1 \, \mathrm{K}$  offset applied, the parcel method tends to diagnose an MLH about  $150\text{--}200 \, \mathrm{m}$  higher on average. The bulk Richardson method only produces higher MLH values in cases of very high wind speed. During morning growth, the offset parcel method is also more likely to select the top of a residual layer, producing larger outliers ( $300\text{--}1000 \, \mathrm{m}$ ) that raise the mean difference to about  $250 \, \mathrm{m}$  when considering all radiosonde launches.





For the MPD-aerosol and radiosonde bulk Richardson method comparisons, the strongest agreement is in the total comparison ( $R^2 = 0.919$ , bias =  $-160 \,\mathrm{m}$ ). However, the MPD-aerosol method occasionally tracked morning residual layers. These false identifications are evident in lidar data where the retrieved MLH sinks during morning growth, which is inconsistent with surface-driven development. Similarly, some radiosonde profiles show that the bulk Richardson method failed to trigger on weak inversions, resulting in unrealistically deep MLH estimates. In the clear-sky afternoon comparison (Panel (f)), where residual layer misidentification is not a factor, correlation decreases slightly ( $R^2 = 0.819$ ; bias =  $-316 \,\mathrm{m}$ ).

While these statistics could suggest that the MPD-aerosol method underestimates the MLH, it is also important to consider that the radiosonde may not be spatially aligned with the lidar. Figure 5 shows an example where the radiosonde bulk Richardson method retrieves a higher MLH than the MPD-aerosol method. Here, the MPD-aerosol method identifies a strong aerosol gradient at 3885 m (line 4), visible in the aerosol backscatter coefficient profile. The radiosonde profiles also show this transition, visible in the increase in the virtual potential temperature and a sharp decrease in the relative humidity at the same height. However, the bulk Richardson number stays below the 0.25 threshold and instead returns an MLH at the next inversion at 5039 m (line 2), over 1 km higher. The offset parcel (5121 m, line 1) and MPD-thermodynamic (4657 m, line 3) methods also select higher altitudes than the MPD-aerosol, though the upward shift in the MPD-thermodynamic retrieval may reflect the bias introduced by the 1 K offset.

**Figure 5.** Radiosonde and MPD profiles for the radiosonde launched at 22:15 UTC on 26 July 2023. The figure shows (a) virtual potential temperature from the radiosonde and MPD, (b) aerosol backscatter coefficient from the MPD, (c) relative humidity from the radiosonde and MPD, and (d) bulk Richardson number from the radiosonde. Horizontal dotted lines indicate MLH retrievals: 1—Radiosonde offset parcel method, 2—Radiosonde bulk Richardson method, 3—MPD-thermodynamic method, and 4—MPD-aerosol method. The solid black lines represent radiosonde profiles in panels (a), (c), and (d), and the dashed blue lines represent MPD profiles in panels (a), (b), and (c). All heights are AGL.

The radiosonde likely ascended within a localized updraft or followed shortly behind one, sampling air that had been lofted above the transition layer. Between 3.6 and  $4.3 \, \mathrm{km}$ , the radiosonde's ascent rate was 1 to  $2.8 \, \mathrm{ms^{-1}}$  above its expected value, indicating vertical air motion consistent with an updraft. This interpretation is also supported by the absence of elevated aerosol backscatter or relative humidity in MPD data above  $4 \, \mathrm{km}$ , despite elevated humidity in the radiosonde data. By the time the radiosonde reached  $4 \, \mathrm{km}$ , strong winds had carried it  $6.4 \, \mathrm{km}$  to the northeast. The radiosonde sampled a weak inversion at  $3.9 \, \mathrm{km}$ , with a near-adiabatic layer above  $(4.1–5 \, \mathrm{km})$  capped by a stronger inversion. The MPD resolved only one strong inversion, possibly due to challenging conditions (high-altitude MLH and strong solar background) or because only one inversion was present over the site. This day was hot  $(36 \, ^{\circ}\mathrm{C})$ , extremely dry (11%), and had vigorous fair-weather convection (convective velocity scale  $\approx 3.5 \, \mathrm{ms^{-1}}$ ). DWL data showed an intense updraft 30 minutes after launch, with a peak velocity of  $9.8 \, \mathrm{ms^{-1}}$  near  $2.5 \, \mathrm{km}$ . Such strong updrafts frequently overshoot the transition layer, injecting well-mixed air into the layer above (see Sullivan et al. (1998)). The MPD observed thin clouds an hour later between  $4.1 \, \mathrm{and} \, 4.5 \, \mathrm{km}$ , suggesting an updraft overshot the inversion and reached the condensation level. The radiosonde could have encountered a similar updraft or the remnants of one, shifting the MLH higher than the one observed by the MPD-aerosol method.






As in Section 5.1.2, the MLH definition becomes important. If the MLH is defined as the height of the transition layer that caps the mixed layer, 3.9 km is most appropriate. However, elevated virtual potential temperature and relative humidity in the radiosonde data, along with later cloud formation observed by the MPD, suggest occasional deeper mixing, with a near-adiabatic layer extending to 5 km, consistent with the bulk Richardson retrieval. If the MLH is instead defined by the vertical extent of surface influence, this deeper height is more appropriate, especially when considering the air mass sampled by the radiosonde.

The radiosonde comparisons generally support the validity of both the MPD-thermodynamic and MPD-aerosol methods, assuming their underlying assumptions are met. They also highlight the limitations of the MPD methods, especially in cloudy conditions. At the same time, they expose limitations in using radiosondes as a validation reference. For example, on clear-sky afternoons, radiosondes often retrieved higher MLH values than the MPD-aerosol method. This discrepancy may result from radiosondes ascending through localized updrafts, yielding higher MLHs than those observed directly over the MPD. These limitations, along with inconsistent MLH definitions across methods, must be considered when interpreting results. Nonetheless, radiosondes remain one of the most reliable tools for determining the MLH due to their ability to operate in both clear and cloudy conditions. In contrast, lidars are limited by cloud cover and precipitation, but can monitor the temporal resolution, which can help determine whether an identified layer evolves consistently with the conceptual model of the MLH.

To evaluate whether the MPD-aerosol method could operate independently of model input, a self-contained version was tested using a top limiter derived from MPD virtual potential temperature profiles instead of the HRRR model. The limiter was defined as the last altitude where the virtual potential temperature equaled the surface value +3 K. Performance was slightly degraded. In these analyses (not shown), the  $R^2$  values for the total, clear sky, and clear sky afternoon were 0.747, 0.838, and 0.792 with biases of -100 m, -65 m, and -314 m, respectively.

# 480 5.3 Series Comparisons


This section presents mean time series comparisons of the MPD-aerosol method, MPD-thermodynamic method, HRRR model, and DWL method to assess retrieval behavior across the full diurnal cycle for the  $M^2HATS$  dataset. A normalized time scale (0 = sunrise, 1 = sunset) enables direct comparison across days with varying lengths (ranging from 12.5 to 14.25 hours). Figure 6 shows the diurnal MLH evolution for each method. Data were grouped into ten bins (width 0.1, centered at 0.05, 0.15, ..., 0.95), with mean MLH and  $\pm 1\sigma$  error bars plotted. This corresponds to a spacing of about 75–85 minutes, depending on day length. Only clear-sky cases were included, though sparse cloud cover was permitted if none were directly above the MPD. Because the MPD-aerosol method only operates between sunrise and sunset, all comparisons were limited to this daytime period. After excluding the two days affected by Hurricane Hilary, the time series comparison included 50 days for the MPD and HRRR methods. For DWL comparisons, an additional 17 days with signal dropouts were excluded, leaving 33 days.

These plots are not statistical averages for the M<sup>2</sup>HATS campaign. Filtering cloudy data raises the average MLH values, and comparisons involving the DWL are biased low due to the exclusion of days with the deepest PBLs.

**Figure 6.** Mean MLH time series from M<sup>2</sup>HATS under clear-sky conditions, shown as a function of normalized time with standard deviation error bars. The figure shows (a) MPD-aerosol compared with DWL, (b) MPD-aerosol compared with HRRR (default), (c) MPD-aerosol compared with MPD-thermodynamic, (d) MPD-thermodynamic compared with DWL, (e) MPD-thermodynamic compared with HRRR (default), and (f) MPD-thermodynamic compared with HRRR (parcel with 1 K offset). All heights are AGL.

The left column of Figure 6 compares the MPD-aerosol and MPD-thermodynamic MLH retrievals to the DWL. In the morning, at normalized time 0.15, the MPD-aerosol method is 420 m (158%) higher than the DWL, and the MPD-thermodynamic method is 160 m (58%) higher. During the core of the day (0.35-0.85), the aerosol method remains 50-540 m (4-27%) above

the DWL, while the thermodynamic method is 320–730 m (28–35%) higher. At 0.95 (evening transition), the aerosol method diverges by 810 m (52%) higher than the DWL, while the thermodynamic method shows only a small bias of 15 m (1%).

The MPD-aerosol method diverges from the DWL at the start and end of the day because aerosol stratification is often an unreliable tracer for the MLH during periods of weak turbulence. In the morning, residual layers and low-altitude aerosol stratification result in an upward bias, while in the evening, the MPD-aerosol method remains lofted while the DWL retrieval decreases (as discussed in Section 5.1.1). The thermodynamic method more closely follows the DWL shape but remains consistently higher, likely due to the 1 K offset and differing detection criteria. These differences are consistent with previous studies. For example, Schween et al. (2014) found that DWL MLH estimates were systematically lower than those from an aerosol lidar using a similar wavelet approach. Krishnamurthy et al. (2021) reported that DWL retrievals underestimated the MLH relative to radiosonde profiles, and Smith and Carlin (2024) observed a consistent low bias using a fuzzy logic algorithm that relied heavily on DWL data.







This bias likely reflects differences in how each method defines and detects the MLH, whether by turbulence, passive tracers, or thermodynamic structure. The DWL tracks the mixing height rather than the MLH based on thermodynamic or aerosol structure. When turbulence does not reach the top of the mixed layer, the DWL will yield lower estimates than the MPD methods, which are sensitive to buoyancy and aerosol gradients even in the absence of turbulence. This behavior appeared in the 8 September case (Section 5.1.2), where the MPD retrievals followed the entrainment of a near-adiabatic, aerosol-rich plume at the top of the mixed layer while the DWL estimates remained lower. Although that case was an extreme example, aerosols and well-mixed air often extend above the turbulent layer to a lesser extent on typical convective days. Additionally, the DWL's bottom-up approach may favor lower indicators for the MLH than the MPD-based approaches.

The middle column of Figure 6 compares the MPD-aerosol and MPD-thermodynamic MLH retrievals to the default HRRR model. Between normalized times 0.35 and 0.85, the MPD-aerosol method agrees closely with the HRRR PBLH, with a maximum bias of  $-190 \,\mathrm{m}$  (8%) at 0.55. Outside this period, it shows an upward bias of  $+391 \,\mathrm{m}$  (145%) in the morning (0.15) and  $+750 \,\mathrm{m}$  (35%) in the evening (0.95). The MPD-thermodynamic method is biased high at 0.35 ( $+440 \,\mathrm{m}$ , 36%) but aligns better during the transition periods, with a morning bias of  $+150 \,\mathrm{m}$  (55%) and an evening bias of  $-380 \,\mathrm{m}$  (18%).

The bottom-right panel compares the MPD-thermodynamic method to the HRRR model using the offset parcel method, which removes bias from differing diagnostic definitions. The agreement improves slightly. The largest difference is  $-430 \,\mathrm{m}$  (13%) at normalized time 0.55, with biases of  $+130 \,\mathrm{m}$  (43%) at 0.15 and  $-760 \,\mathrm{m}$  (30%) at 0.95. The top-right panel directly compares the MPD-aerosol and MPD-thermodynamic methods. The aerosol method is higher in the morning and evening due to aerosol stratification and residual layers but lower during the middle of the day. At normalized times 0.15, 0.45, and 0.95, its bias relative to the thermodynamic method is  $+240 \,\mathrm{m}$  (58%),  $-690 \,\mathrm{m}$  (28%), and  $+1130 \,\mathrm{m}$  (64%), respectively.

In summary, the MPD-aerosol method performs reliably from late morning to evening in convective conditions, approximately from 0.35 and 0.85 normalized time for this study. It shows small biases compared to the default HRRR model during this period. However, it is consistently higher than DWL MLH retrievals and lower than MPD-thermodynamic retrievals, likely due to diagnostic differences. Outside this window, aerosol stratification is an unreliable MLH tracer. The MPD-thermodynamic method, on the other hand, tracks the MLH evolution throughout the day without the same limitations caused by using aerosols.

However, the offset parcel method introduces a consistent upward bias relative to the other retrievals. The MPD-thermodynamic method shows closer agreement to the offset parcel MLH retrieval applied to the HRRR profiles than any other method. While reducing the offset to minimize this bias may seem beneficial, doing so increased false MLH detections due to noise. Notably, the two MPD methods perform best under opposite conditions, suggesting the potential for a complementary approach. However, it remains unclear how their diagnostic differences could be reconciled into a continuous MLH retrieval.

## 535 6 Discussions






The MPD-aerosol method accurately tracks the aerosol layers with high temporal and vertical resolution. It shows strong agreement with radiosondes, with  $R^2$  values ranging from 0.819 to 0.919, depending on the subset. The MPD-aerosol method performed well in time series comparisons with the DWL and HRRR model under convective conditions, specifically between 0.35 and 0.85 normalized time. Outside this range, performance degrades due to limitations in aerosol structure. Before 0.35, the method occasionally misidentifies low-altitude residual layers as the MLH, even with a top limiting constraint. After 0.85, it tends to remain at the residual layer instead of detecting the newly forming stable layer, as the top of this layer rarely exhibits a strong enough aerosol gradient. These limitations reflect the broader challenge of aerosol stratification becoming an unreliable tracer of the MLH during weak turbulence.

The MPD-thermodynamic method the other hand, has fewer issues during the early morning and evening transitions and shows closer agreement with the HRRR model during these periods. However, its lower resolution limits its ability to track dynamic MLH changes with the precision of the aerosol method. Second, it has a significant bias in cloudy conditions, often falsely identifying an MLH beneath cloud layers. Third, noise in the temperature profiles requires an offset in the parcel method, shifting the retrieved MLH upward. When compared to radiosondes or the HRRR model using the same offset parcel method, it shows reasonable agreement.  $R^2$  values exceed 0.66 for all radiosonde comparisons. However, the offset complicates model evaluation, requiring post-processing with a matching MLH retrieval.

The M<sup>2</sup>HATS campaign was the first full-length deployment of the MPD's temperature-profiling capabilities. Hardware upgrades are expected to reduce the temperature bias reported by Hayman et al. (2024), and data processing improvements may enhance performance in cloudy scenes, a known limitation. These improvements may also benefit the MPD-aerosol method, as improved virtual potential temperature profiles could allow the top limiter to be constrained using MPD data alone, with performance comparable to using HRRR data. In addition to the continuous improvements to hardware and processing, other potential advancements to the MPD data products include the retrieval of variables relevant to convection (e.g., Hoffman and Demoz, 2025), such as equivalent virtual potential temperature, CAPE (convectively available potential energy), and CIN (convective inhibition).

To test the generalizability of the algorithms beyond the M<sup>2</sup>HATS environment, the MPD-aerosol and MPD-thermodynamic methods were applied to the Boulder dataset. As shown in Figure 7, the algorithms reproduced similar diurnal MLH patterns and showed typical PBL evolution on the first and fourth days with some disagreements between methods on the second and third. On the second day, aerosols did not extend through the full thermodynamic MLH, leading to lower estimates by the

MPD-aerosol method. On the third, the MPD-thermodynamic method produced MLH estimates that were unexpectedly low during the afternoon and appear to be influenced by a persistent instrument bias near 1 km, visible as a horizontal band in the potential temperature field. These results demonstrate that the methods are not overfitted to the unique conditions of the  $M^2$ HATS campaign and suggest they can be applied more broadly.




**Figure 7.** Time series of MLH retrievals for a dataset from 9–12 March 2025 in Boulder, Colorado, USA. **Top:** Aerosol backscatter coefficient at 770 nm, overlaid with MLH retrievals from the MPD-aerosol, HRRR MLH estimates, and cloud base heights. **Bottom:** Difference between atmospheric and surface virtual potential temperature overlaid with MPD-thermodynamic, HRRR MLH estimates, and cloud base heights. Marker density is downsampled for readability and does not reflect the native temporal resolution of each method. Vertical dashed lines indicate sunrise and sunset times. All heights are AGL.

Additional limitations were identified in the other MLH retrieval methods. While radiosondes remain among the most trusted tools for MLH retrievals, they are not without limitations. In convective conditions, the bulk Richardson and offset parcel methods located higher MLHs. This tendency may reflect deeper mixing, but can also result from radiosondes drifting into different air masses or encountering localized updrafts, complicating their use as validation references. Nonetheless, radiosondes offer high vertical resolution and are a mature technology, making them a valuable benchmark. The DWL also has limitations. It consistently returned lower MLH values than the other methods, largely because it detects the height of active turbulence (i.e., the mixing height) rather than relying on aerosol stratification or thermodynamic structure. Its bottom-up retrieval approach may also favor lower indicators of the MLH. Though data dropouts occurred on some days, when the signal reached the top of the mixed layer, the DWL retrievals had the highest vertical and temporal resolution of any remote sensing method tested.

Together, these results demonstrate that no single instrument can reliably monitor the MLH under all conditions and that every method has strengths and weaknesses. Further, they highlight that the MLH can be poorly defined in non-canonical PBL development, and differences between methods should be expected in such cases. These challenges underscore the value of

synergistic approaches that combine MLH estimates from multiple instruments (e.g., Duncan Jr. et al., 2022; Smith and Carlin, 2024; Zhang et al., 2025), especially when one system's strengths offset another's weaknesses. For example, the MPD may be complemented by a microwave radiometer, which can retrieve stable boundary layer heights at night within the MPD's blind spot, or a radar wind profiler, which can retrieve the MLH in cloudy or precipitating conditions. At present, the MPD-aerosol and MPD-thermodynamic methods complement each other as agreement between them increases confidence in the retrieved MLH, while disagreement can serve as a useful flag for questionable or unreliable data.

There is increasing interest in applying machine learning to MLH retrievals, either to improve lidar results (e.g., Krishnamurthy et al., 2021; Rieutord et al., 2021; Liu et al., 2022; Wijnands et al., 2024), estimate the MLH from surface observations using lidar-trained models (e.g., Molero et al., 2022; Peng et al., 2023; Su and Zhang, 2024; Stapleton et al., 2025), or determine a best estimate by combining multiple MLH retrievals obtained from different instruments and methods (e.g., Zhang et al., 2025). A central challenge for these approaches is the lack of ground truth. Models inevitably inherit the limitations and assumptions of the methods used to generate their training data. MPD-aerosol or MPD-thermodynamic retrievals could provide training data under favorable conditions. These retrievals, along with their associated aerosol and thermodynamic profiles, may also serve as inputs to future machine learning models.

There is also a growing effort to monitor the MLH and mixing height at national and continental scales through networks of low-cost instruments like micropulse lidars (Lewis et al., 2013; Su et al., 2020; Roldán-Henao et al., 2024) and ceilometers (Haeffelin et al., 2012; Caicedo et al., 2020; Kotthaus et al., 2020), which support widespread deployment. Although the MPD is likely to be more expensive than ceilometers or MicroPulse lidars, it is also well suited for network-based applications. A network of MPDs could provide temperature and humidity profiles, with MLH estimates and aerosol profiles as additional data products.

## 7 Conclusions





This study evaluated the performance of two automatic MLH retrieval algorithms, one using the MPD's thermodynamic profiling capabilities and the other using its aerosol backscatter profiling capabilities. The methods were compared to each other and against MLH retrievals from radiosondes, a DWL, and the HRRR model. The MPD-aerosol method performed well during the day in convective conditions but struggled when the aerosol gradients decoupled from the MLH in the morning and evening. The MPD-thermodynamic method avoided some of these issues but was limited by coarser temporal and vertical resolution, sensitivity to clouds, and an offset required to manage instrument noise that introduced an upward bias. The two methods are complementary. When the methods agree, confidence in the MLH retrievals increases. With appropriate quality control and awareness of limitations, the MPD-aerosol and MPD-thermodynamic methods can provide valuable MLH retrievals for boundary layer research, model evaluation, and air quality studies.

This work also highlights how different diagnostic definitions of the MLH can produce different results. Although it is generally accepted that aerosol gradient methods follow residual layers during the evening transition, leading to discrepancies with thermodynamic or turbulence-based definitions, such differences are not limited to those cases. For example, during

morning growth periods, thermodynamic parcel methods often produce step-like jumps in the MLH when residual layers are near-adiabatic, whereas aerosol and turbulence-based methods indicate more gradual growth. These discrepancies underscore the importance of clearly defining the MLH and carefully selecting a diagnostic approach suited to the specific scientific objective.

## **Appendix A: Mathematical Description of the HWT**


The HWT forms the basis of the MPD-aerosol MLH retrieval algorithm. The Haar wavelet is an antisymmetric square wave function defined by Brooks (2003) as:

$$h\left(\frac{u-z}{a}\right) = \begin{cases} 1 & \text{if } b - \frac{a}{2} \le z 

Figure B1. Example virtual potential temperature profiles from the MPD (black) and radiosonde (red dashed line, shown for reference). Vertical dotted lines indicate the surface value ( $\theta_{v,\text{surf}}$ ), the +1 K offset, and the +3 K top limiter. The algorithm starts at the lowest available level (1), searches upward to the +3 K intersection (2), then searches downward to the first level at or below the +1 K offset (3). The MLH is diagnosed as the last level before (3). Cyan circles mark the three reference points: (1) lowest available level, (2) first +3 K intersection, and (3) first level at or below +1 K in the downward search.

Figure B1 illustrates the application of these thresholds to a representative virtual potential temperature profile, with a colocated radiosonde profile included only for reference. Fluctuations in  $\theta_v$  within the mixed layer, as seen between 800 m and 2800 m, are physically implausible and not observed in the radiosonde since such variations would immediately trigger updrafts. They can appear in practice due to noise in the retrievals. The empirical thresholds mitigate against these effects. Applying the parcel method to another remote-sensing instrument requires considering its noise characteristics and selecting appropriate values for the offset and top limiter. The offset should be larger than typical instrument bias to avoid false detections, and the top limiter should be high enough to lie above the mixed layer but low enough to filter spurious high-altitude noise. A smaller offset will likely lead to more similar results when compared to other MLH diagnostic methods (e.g., the bulk Richardson method). However, the exact offset value is less important than ensuring that the same diagnostic definition is used across instruments and models (LeMone et al., 2013; Chu et al., 2019).

# Appendix C: The Bulk Richardson Radiosonde Method



The bulk Richardson number method relies on radiosonde profiles of wind speed, wind direction, and virtual potential temperature, together with surface observations. The potential temperature is given in Equation B1, and the inputs are provided by the radiosonde.

The virtual potential temperature equation requires the specific humidity, so the relative humidity measured by the radiosonde must be converted to the specific humidity. This requires an intermediate calculation of the saturation vapor pressure and the mixing ratio (the mass ratio of water vapor to dry air). The saturation vapor pressure in hPa as a function of height AGL, z, can be found using the Tetens equation (following Murray, 1967).

$$e_s(z) = \begin{cases} 6.1078 \cdot \exp\left(\frac{17.27 \cdot (T(z) - 273.15)}{T(z) - 35.85}\right) & \text{if } T(z) \ge 273.15\\ 6.1078 \cdot \exp\left(\frac{21.875 \cdot (T(z) - 273.15)}{T(z) - 7.65}\right) & \text{if } T(z) < 273.15 \end{cases}$$
(C1)

Once the saturation vapor pressure is found, the actual vapor pressure can be calculated as

$$e(z) = \frac{RH(z)}{100} \cdot e_s(z),\tag{C2}$$

where RH(z) is the relative humidity measured by the sonde. The mixing ratio, which is the ratio of water vapor mass to dry air mass, can then be obtained from the ratio of water vapor pressure to dry air pressure, using the ideal gas law as

$$r(z) = \frac{\epsilon \cdot e(z)}{p(z) - e(z)},\tag{C3}$$

where  $\epsilon$  is the ratio of the molecular weight of water vapor to the molecular weight of dry air, which is 0.622. The specific humidity can be found as

$$q(z) = \frac{r(z)}{1 + r(z)},$$
 (C4)

and the virtual potential temperature can be found using Equation B4. The bulk Richardson number is

$$Ri_b(z) = \frac{\frac{g}{\theta_{v,surf}} \left[ \theta_v(z) - \theta_{v,surf} \right] (z - z_{surf})}{\left[ u(z) - u_{surf} \right]^2 + \left[ v(z) - v_{surf} \right]^2},$$
(C5)

where g is gravitational acceleration, u(z) and v(z) are horizontal wind components, and the subscript "surf" represents the surface observations.

To interpret the bulk Richardson number physically, it is helpful to examine it as a ratio of the stability term in the numerator and the shear generation term in the denominator. The numerator increases with static stability (a positive difference in  $\theta_v$  with respect to the surface). Also note that the numerator is zero when  $\theta_v(z) = \theta_{v,surf}$ . The denominator represents the kinetic energy available from vertical wind shear to generate turbulence. For negative bulk Richardson numbers, the atmosphere is unstable and the flow is dominated by turbulent mixing. This is typical within the mixed layer. For large positive bulk Richardson numbers, the flow is dominated by static stability, leading to the suppression of turbulence. For small positive bulk Richardson numbers, the kinetic energy from wind shear is able to create turbulence despite the weak static stability. Several criteria have been suggested for a critical bulk Richardson number that marks the transition from an atmosphere that sustains turbulence to one that suppresses it. The MLH is diagnosed as the first height where  $Ri_b(z) \ge Ri_{b,crit}$ . For this study, a critical bulk Richardson number of  $Ri_{b,crit} = 0.25$  is used, following Seibert et al. (2000), Seidel et al. (2012), and Guo et al. (2021).

Because the surface-level radiosonde observations are often unreliable due to handling during launch.  $\theta_{v,vert}$  is taken from

Because the surface-level radiosonde observations are often unreliable due to handling during launch,  $\theta_{v,surf}$  is taken from the first valid in-air sonde level, typically about 10 m AGL. The initial wind measurements are likewise uncertain because the radiosonde swings below the balloon during the initial ascent. To avoid this contamination, the surface wind components  $u_{surf}$  and  $v_{surf}$  are substituted with co-located measurements from the ISS weather station's anemometer at the time of launch.

# Appendix D: Determining the Vertical Velocity Variance Threshold

The appropriate threshold relies on an empirical equation from mixed layer similarity theory, which relates the vertical velocity variance to the MLH. This equation, first proposed by Lenschow et al. (1980), is:

$$\frac{\overline{w^2(z)}}{w_*^2} = 1.8 \left(\frac{z}{z_i}\right)^{2/3} \left(1 - 0.8 \frac{z}{z_i}\right)^2 \tag{D1}$$

where  $\overline{w^2(z)}$  is the mean squared vertical velocity as a function of height AGL, z. Assuming a mean vertical velocity of zero, then  $\overline{w^2(z)} = \sigma_w^2(z)$ , where  $\sigma_w^2(z)$  is the vertical velocity variance. The term  $w_*$  is the convective velocity scale, and  $\frac{z}{z_i}$  represents the normalized height, where  $z_i$  is the MLH. Rearranging, setting  $z = z_i$ , and substituting terms yields

$$\sigma_w^2(z_i) = 0.072w_*^2. \tag{D2}$$

The convective velocity scale is defined as:

$$w_*^2 = \left(\frac{g}{\theta_{v,r}} \overline{w'\theta'_{v,s}} z_i\right) \tag{D3}$$

where  $\theta_v, r$  is the mean virtual potential temperature within the mixed layer and  $\overline{w'\theta'_{v,s}}$  is the surface buoyancy flux. Since the T30 ISFS array directly measures surface buoyancy flux, a dynamically varying vertical velocity variance threshold can be used.

An iterative approach is used: The threshold for  $\sigma_w^2$  is set to  $0.162~\mathrm{m^2\,s^{-2}}$  ( $w_* = 1.5~\mathrm{m\,s^{-1}}$ ), and a first estimate of  $z_i$  is obtained. This estimate, along with the measured buoyancy flux and surface virtual potential temperature  $\theta_{v,r}$ , was used to compute an updated convective velocity scale and a new  $\sigma_w^2$  threshold. If this updated threshold falls below  $0.072~\mathrm{m^2\,s^{-2}}$  ( $w_* = 1~\mathrm{m\,s^{-1}}$ ), it was set to  $0.072~\mathrm{m^2\,s^{-2}}$  to prevent high MLH estimates during weak convection. Then, a new  $z_i$  was found, and the process was repeated until convergence. The surface buoyancy flux was taken as the average 3 m value from 17 ISFS towers. This method accounts for changes in surface forcing. However, it is not entirely satisfying, as the mixed layer similarity relationship assumes a fully developed mixed layer, typically only present from midday to the evening transition. Nonetheless, this approach provides a more physically grounded estimate of the MLH than an approach that uses a fixed threshold.

Code availability. Relevant code can be found at https://github.com/ColbergMontanaStateUniversity/mlh-amt2025.git (Colberg, 2025). Per-740 ceptually uniform colormaps are used in this project (Crameri et al., 2020).

Data availability. All datasets used in this study are publicly available. The data from M<sup>2</sup>HATS are accessible at https://www.eol.ucar.edu/field\_projects/m2hats (NCAR, 2023). MPD data collected in Boulder are accessible at https://data.eol.ucar.edu/dataset/100.034 (NCAR, 2025).

Author contributions. LC performed the formal analysis, developed the code, and wrote the manuscript under the supervision of KSR. KSR,
 SMS, MH, and LC conceptualized the study. SMS, MH, and RAS collected the data. MH processed the MPD temperature and humidity measurements. All authors contributed to reviewing and editing the manuscript.

Competing interests. The authors declare that they have no conflicts of interest.



Acknowledgements. The authors thank Dr. Shane Mayor for productive conversations that helped guide the development of this work. This research was supported in part by National Science Foundation Grant No. 2234047 and EPSCoR Cooperative Agreement OIA-2242802. The M<sup>2</sup>HATS campaign was funded under National Science Foundation Grant Nos. 2054983 and 2054969. This material is based upon work supported by the National Center for Atmospheric Research, which is a major facility sponsored by the National Science Foundation under

| Cooperative Agreement No. 1852977. Any opinions, findings, and conclusions or recommendations expressed in this material are t | hose of |
|--------------------------------------------------------------------------------------------------------------------------------|---------|
| the authors and do not necessarily reflect the views of the National Science Foundation.                                       |         |

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
