# Peer review of "Mixed Layer Height Retrievals Using MicroPulse Differential Absorption Lidar"

_EGUsphere, 2025_

## Referee Comment (RC1)

The manuscript is well-written, well-structured, and presents detailed investigation of mixed layer height (MLH) detection using mainly MicroPulse DIAL (MPD) lidar measurements, but also radiosondes, Doppler Wind Lidar and MRRR model. The authors have created nice and informative figures. They evaluate two retrieval approaches — one aerosol-based and one thermodynamically based — while also provide meaningful discussion on the different definitions of the planetary boundary layer (PBL), the diversity of retrieval techniques, and the challenges associated with their comparison. In my opinion, the work falls well within the scope of the journal and merits publication after some revisions.

This is a difficult topic, given the inherent challenge of comparing results from different instruments and retrieval algorithms. I find that the paper succeeds in providing well-supported assessment of MLH using multiple approaches with a focus on MPD observations. Nevertheless, there are several methodological aspects that in my opinion require further clarification to improve reproducibility and ensure the work can be applied in future studies.

**Major comment:**
The thermodynamic retrieval of MLH from lidar data using the parcel method is an interesting and novel contribution. However, as described in the manuscript, it appears to rely on several empirically set thresholds. The authors should elaborate further on the rationale for these thresholds and provide enough detail to allow the parcel method to be applied robustly to other lidar datasets.

**Other comments/questions:**

1. The abstract would benefit from specifying the locations where the measurements were conducted. This information is important for readers to interpret the environmental context of the results (for example it is not at the desert or a coastal site). Additionally, please clarify at the abstract whether the analysis was restricted to cloud-free condition.

2. Just a comment (no need for revisions): The introduction effectively emphasizes that, ultimately, the definition for the boundary layer is critical to ensure that comparisons make sense.

3. Lines 50–51: I suggest softening this statement. In real-world lidar measurements, it is common to observe a residual layer above the mixing layer. While the residual layer may not be part of the PBL in practice, it is often included in the conceptual model definition. This nuance is worth acknowledging to avoid oversimplifying the relationship between MLH and PBLH.

4. Lines 41-49: The authors give a solid overview of lidar-based PBLH retrievals and their challenges. Since your work deals with comparing and evaluating different methods and synergies, you might also briefly acknowledge recent efforts that use multi-sensor approaches (e.g., combining lidars, radiosondes, microwave radiometers, and/or models) to help reconcile differences in estimates from various techniques and synergies (Moreira et al., 2019, Tsikoudi et al., 2022, Chen et al.,

2022, and Zhang et al., 2025). Including a short sentence and these references would broaden the context for the study, like at the discussion (lines 497-499).

5. Line 54: Please provide a reference for the HRRR model here. You might consider repeating the Dowell et al. (2022) citation already used in line 203.

6. Section 2: Since aerosols are used as passive tracers in this study, I suggest adding a separate paragraph (or integrating into the existing text) discussing how they mix and exist in all the layers mention here. Specifically, maybe mention that aerosols are often trapped within the boundary layer and residual layers, and lidar measurements of aerosol backscatter allow us to detect these layers and define their tops. While this section is primarily theoretical and has been widely discussed in many previous studies, including this aerosol information could help readers better understand how lidar-based PBL detection works.

7. Fig. 1: Please define the abbreviation "AGL" as "above ground level". If I am not mistaken, this is the first occurrence of the abbreviation in the manuscript.

8. Lines 75–77: I suggest noting that the residual layer is detached from surface properties, and while some may argue it is not strictly part of the PBL, at the end of the day its inclusion depends on the definition used. It could also be useful to cite studies that have investigated the residual layer using lidar, for example Fochesatto et al., 2001.

9. Line 87: The term "transition layer" is generally difficult (if not impossible) to capture by lidars due to overlap effects, which is also the case for MPD measurements.

10. Lines 91–98: The discussion here makes perfect sense, but could the authors provide references to support these statements?

11. Line 103: From a quick search, it seems that Tonopah is surrounded by several notable mountain ranges and peaks. How can it be described as a "flat basin" given its elevation of 1641 m? Additionally, there is a statement regarding Tonopah's MLH—has this been investigated in other studies or it is an empirical statement? What are the corresponding climate characteristics at the Colorado site? The measurement period seems short and may represent only limited atmospheric conditions. Finally, it would be helpful to clarify the typical aerosol types encountered at these sites (e.g., biomass, pollen, dust), especially because they are used as tracers for the MLH. Both Boulder and Tonopah are located near notable terrain features (e.g., Cheyenne Mountain for Boulder; Lone Mountain and Butler Siebert for Tonopah).
Have the authors considered the impact of complex terrain on their MLH observations? Are there

any previous studies investigating orography effects at these sites? Also, both sites are around 1600 m elevation, but their surface characteristics differ (forested vs. arid); could this influence the measurements or interpretation?

12. Line 115: HSRL measurements still require careful instrument calibration (e.g., molecular channel normalization, overlap correction near the surface) and quality control. It would also be helpful to mention the wavelength of operation.

13. Line 116: How is the virtual potential temperature calculated? Are dry and moist air considered, and are clouds excluded? Is the water vapor mixing ratio used? How is pressure incorporated? This information, along with the relevant formulas, should be described in an appendix. In my opinion, since this is the first time the parcel method is applied to lidar data, a detailed methodological description is necessary.

14. Lines 118–119: When the authors say "*the MPD had a minimum range of 318 m AGL and a vertical resolution of 150 m with 37.5 m range bin spacing,*" do they mean that the MPD averages 4 bins to improve the signal-to-noise ratio (4 × 37.5 m = 150 m)? So, the meaningful profile resolution is 150 m, while the raw data points are spaced every 37.5 m and are noisier? Maybe I am missing something, please clarify.

15. Line 125: I suggest, if possible, to provide references for the Halo Streamline Pro and Vaisala instruments. Additionally, were any corrections applied for noise? What type of scan was performed (horizontal, vertical, or both)?

16. Lines 130–131: Could the authors clarify why precipitation data are needed for the DWL? Does the instrument automatically stop or flag data during rain to avoid contamination, or is this used to filter MLH retrievals?

17. Line 148: How are cloudy data removed from the lidar observations? Does the algorithm apply a threshold on the raw signal? Removing clouds is critical in lidar studies, and it would be helpful to clarify how the method distinguishes clouds from dense aerosol layers, which can also produce strong backscatter. Lines 163 and 186–187 suggest that some cloud-affected measurements remain in the dataset; please elaborate on how these are treated or flagged.

18. Appendix A – Is b the translation parameter? Please clarify, it may be confusing for the reader. I understand that $r_b$ and $r_t$ correspond to the bottom and top of the integration, but the role of b should be explicitly described. How is b defined within the piecewise function?

19. Line 152: If you are referring to Dijkstra (1959), please cite it directly, even if it is already mentioned in de Bruine et al. (2017). This makes it easier for readers who encounter the algorithm for the first time. I think it is a pretty common algorithm in informatics, but not really in atmospheric physics. Also, could the authors clarify whether Pathfinder is a software package used for retrieving the MLH, or if the algorithm was implemented from scratch? Was it applied to 5-minute backscatter profiles? If yes (as mentioned in L53) was it attenuated backscatter? These details will help better understand the methodology.

20. Lines 156–159: The range for the top limiter seems quite large. Is it empirically set? Also, how does the algorithm handle cases where aerosol layers exist close to the top of the mixing layer, or when strong horizontal winds enhance turbulence and lead to unusually high MLH values?

21. Line 163: Again, it is unclear how low clouds are identified and removed from the lidar data. Are specific thresholds applied to the signal? How are these distinguished from dense aerosol layers that could produce similarly strong backscatter? A clear description is needed.

22. Line 176: Please clarify whether this weather station is the same as the one mentioned in line 130. If it is integrated into the MPD system, this should be also stated in Section 3.3.

23. Lines 178–187: It seems that many of these thresholds and limits were set empirically. A visual representation (e.g., a diagram or flowchart) would help the reader understand the workflow. This is especially useful for someone applying the thermodynamical parcel method to their own lidar data. For instance, a figure showing how the virtual potential temperature is calculated, including how different time resolutions for humidity and temperature are handled, would make the method more transparent. I would suggest, if not included in the main text, consider adding it to the appendix.

24. Line 193: The parcel method using a 1 K offset from the surface value, is applied in the same way as for the MPD? So that the initial condition or reference for both methods is consistent (or, if different, how it differs).

25. Lines 194–201: The comparison between radiosonde–Richardson and MPD–aerosol MLH retrievals is reasonable for daytime convective conditions, but it can diverge in transition periods or multi-layer situations. After sunset or in late morning, elevated well-mixed layers or residual aerosol layers can persist aloft, so the Richardson method (sensitive to turbulence) may detect a different top than the aerosol-based MLH. The authors partly mention this (L202–203), but it should be emphasized that their comparison is valid for daytime convective periods, as the algorithm does not run at night. It would also help to specify until what local time the algorithm provides retrievals. Is it 20:00 local time? Additionally, in my opinion it is worth expanding on the Richardson method in the appendix, similar to the treatment for Wavelet Haar and the wind lidar, to clarify its application.

26. Lines 220–221: It should be briefly clarified how virga was manually removed from the lidar data. For example, was a threshold applied to the attenuated backscatter or signal strength? Providing

this detail helps readers understand how aerosol layers were distinguished from falling precipitation or virga.

27. Lines 228–229: Could you clarify whether the parcel method was also applied to the HRRR virtual potential temperature fields? If so, this strengthens the need to include the full description and formulas of the parcel method in the appendix.

28. The authors have done a great job discussing the 6 and 8 September case studies. The manuscript clearly contrasts different boundary layer conditions and aerosol structures, effectively demonstrating how the methodology performs.

29. Section 5.2 is particularly valuable, as it compares the parcel method applied to different instruments—radiosonde (in-situ) and lidar (remote sensing). These instruments measure different tracers for MLH, making the comparison insightful, and the authors describe the results well. However, clarification on collocation is needed: Were the lidar products time-averaged around the radiosonde launch? For example, was the closest 5-min backscatter profile used for the MPD-aerosol method, 10-min water vapor, and 40-min temperature for the MPD-thermodynamic method?

30. L355: Do you mean that the MLH retrieved from the parcel method is on average 250 m higher than the MLH from the Richardson method? If so, why this difference occurs? Both methods detect turbulent layering, but the Richardson method responds directly to shear and turbulence, whereas the parcel method may follow the thermodynamic profile, potentially resulting in systematically higher MLH estimates.

31. L346-349: The description of the different comparisons in each column could be clarified. From the figure, it seems that (a) and (d) show all conditions, (b) and (e) show clear-sky conditions, and (c) and (f) show clear-sky conditions at 15:00 local time. I believe it would help the reader if this were explicitly stated in the text. Also, please clarify how clouds are defined or detected in your MPD retrievals, since this affects which points are included in the clear-sky subsets.

32. Fig 5: Consider maybe adding a small legend for the two lines at the bottom right to make it easier for the reader to distinguish them quickly. Also, using the same solid black line for both radiosonde retrievals (panels a, c) and MPD retrievals (panel b) can be confusing, even if explained in the caption.

33. Section 5.3: It seems that each point in Figure 6 represents roughly 2 hours, given that 0 corresponds to 06:00 LT and 1 to 20:00 LT (14 hours divided by 9 points). Is it true? It would be helpful to clarify the model's time step/resolution, and this information, along with the spatial resolution, should be included in Section 4.5.

---

## Author Comment (AC1)

Response to referee comments

The authors would like to thank the referees for the time and care they devoted to reviewing this manuscript, and for their thoughtful and constructive comments. Their comments are listed below in numerical order, and our responses are provided in **bold blue font.** Line numbers in our responses refer to the revised draft. A marked-up PDF is included, showing the additions in underlined blue font and deletions in .

Reviewer #1

Major Comments

1.  The thermodynamic retrieval of MLH from lidar data using the parcel method is an interesting and novel contribution. However, as described in the manuscript, it appears to rely on several empirically set thresholds. The authors should elaborate further on the rationale for these thresholds and provide enough detail to allow the parcel method to be applied robustly to other lidar datasets.

    **An appendix has been added that describes the MPD-thermodynamic method in detail. It includes:**
    - **The complete derivation of potential temperature, specific humidity, and virtual potential temperature, with all formulas and definitions.**
    - **An explicit description of how the parcel method is applied to MPD profiles, including the treatment of clouds and surface conditions.**
    - **An explanation of the empirically chosen thresholds (1 K offset and 3 K top limiter), their physical rationale, and how they improve the retrievals.**
    - **A figure (Figure B.1) illustrating how the parcel method diagnoses the MLH in a representative virtual potential temperature profile.**
    - **Guidance for applying the method to other remote sensing instruments.**

    **Additionally, the description of the method was made more concise in Section 4.2 as some of the details were moved to the appendix. The new lines (245-256) now read:**

    **"A simple implementation that selects the first altitude where the virtual potential temperature exceeds the surface value is prone to false detections when applied to the MPD data, and high-altitude noise can lead to unrealistically deep MLH estimates. To mitigate these issues, empirical thresholds and a top-down search strategy were used. Full details of the methodology are included in Appendix B. In brief, the MLH is diagnosed with a 1 K offset relative to the surface virtual potential temperature bounded by a 3 K top limiter."**

    **We believe these additions give sufficient rationale for the thresholds and enough detail for the method to be applied to other lidar datasets.**

Other comments/questions

1. The abstract would benefit from specifying the locations where the measurements were conducted. This information is important for readers to interpret the environmental context of the results (for example it is not at the desert or a coastal site). Additionally, please clarify at the abstract whether the analysis was restricted to cloud-free condition.

   **The abstract has been updated to specify the measurement locations and clarify the treatment of clouds. The following text was added:**

   - **Lines 7-8: "using data from the M2HATS field campaign in Tonopah, NV, U.S.A., supplemented by a smaller dataset from Boulder, CO, U.S.A."**
   - **Lines 12-13: "Because lidar generally cannot penetrate clouds, conditions with clouds at or below the MLH are not considered, while those with clouds above the MLH are retained."**

2. Just a comment (no need for revisions): The introduction effectively emphasizes that, ultimately, the definition for the boundary layer is critical to ensure that comparisons make sense.

   **We thank the reviewer for this positive feedback.**

3. Lines 50–51: I suggest softening this statement. In real-world lidar measurements, it is common to observe a residual layer above the mixing layer. While the residual layer may not be part of the PBL in practice, it is often included in the conceptual model definition. This nuance is worth acknowledging to avoid oversimplifying the relationship between MLH and PBLH.

   **The statement was revised to acknowledge complications that can arise in the presence of a residual layer. The new sentence, on lines 55-57, reads:**

   **"This paper presents and evaluates two complementary methods for retrieving the mixed layer height (MLH), which generally corresponds to the PBLH in convective conditions, though the presence of a residual layer, the remnant of the previous day's mixed layer, can complicate the relationship and make the exact MLH dependent on definition. "**

4. Lines 41-49: The authors give a solid overview of lidar-based PBLH retrievals and their challenges. Since your work deals with comparing and evaluating different methods and synergies, you might also briefly acknowledge recent efforts that use multi-sensor approaches (e.g., combining lidars, radiosondes, microwave radiometers, and/or models) to help reconcile differences in estimates from various techniques and synergies

(Moreira et al., 2019, Tsikoudi et al., 2022, Chen et al., 2022, and Zhang et al., 2025). Including a short sentence and these references would broaden the context for the study, like at the discussion (lines 497-499).

**We thank the reviewer for the suggested citations. A sentence was added after the discussion of machine learning to acknowledge multi-instrument approaches. The new sentence is on lines 48-50 and reads:**

**"Other recent research has pursued multi-instrument approaches, including synergistic combinations of instruments and intercomparisons of PBLH estimates from different instruments, to provide a more complete picture of PBL dynamics (e.g., de Arruda Moreira et al., 2018, 2019, 2020; Duncan Jr. et al., 2022; Tsikoudi et al., 2022; Smith and Carlin, 2024; Zhang et al., 2025)."**

**We reviewed Chen et al. (2022) but did not include it here, as it focuses on retrieving the PBLH and other products from a single instrument instead of comparing multiple sensors or methods for finding the PBLH.**

5. Line 54: Please provide a reference for the HRRR model here. You might consider repeating the Dowell et al. (2022) citation already used in line 203.

   **A citation to Dowell et al. (2022) was added on line 61, as suggested.**

6. Section 2: Since aerosols are used as passive tracers in this study, I suggest adding a separate paragraph (or integrating into the existing text) discussing how they mix and exist in all the layers mention here. Specifically, maybe mention that aerosols are often trapped within the boundary layer and residual layers, and lidar measurements of aerosol backscatter allow us to detect these layers and define their tops. While this section is primarily theoretical and has been widely discussed in many previous studies, including this aerosol information could help readers better understand how lidar-based PBL detection works.

   **We have added a new paragraph in Section 2 that explicitly describes aerosol sources, their classification by size, their small settling velocity relative to turbulent motions, their rapid mixing in the mixed layer, their persistence in the residual layer overnight, that aerosols are trapped below the MLH and capping inversion, and that the gradients are detectable by backscatter lidar. The new paragraph is on lines 118-129. It reads:**

   **"An underlying assumption of aerosol-based MLH retrievals is that aerosols act as passive tracers of boundary layer dynamics. We adopt the convention that aerosols are broadly classifiable by size: "nucleation" mode particles (< 0.1 μm), "accumulation" mode particles (0.1–1 μm), and "coarse" mode particles (> 1 μm). Typically, nucleation mode particles are short-lived because they rapidly**

**coagulate into larger particles, and coarse mode particles are likewise short-lived due to efficient gravitational settling. In contrast, accumulation mode particles have a relatively small settling velocity and low coagulation rates, allowing them to persist in the atmosphere for weeks to months, dominating the aerosol population. The relatively small settling velocity of accumulation mode aerosols also means their vertical transport is dominated by turbulent mixing (Pandis et al., 1995). Daytime turbulent motions rapidly homogenize aerosols within the mixed layer, and at night, their weak gravitational settling allows them to remain suspended in the residual layer in the absence of turbulence. Because mixed layer air generally does not penetrate into the free troposphere (Stull, 1988), aerosol concentrations above the MLH or capping inversion are expected to be low, producing sharp gradients in aerosol concentration. Many aerosol-based MLH retrievals rely on identifying these gradients in aerosol lidar profiles (Dang et al., 2019a).”**

7. Fig. 1: Please define the abbreviation "AGL" as "above ground level". If I am not mistaken, this is the first occurrence of the abbreviation in the manuscript.

   **The caption of Figure 1 was revised to define the abbreviation. The updated caption now reads:**

   **"Height is above ground level (AGL)."**

8. Lines 75–77: I suggest noting that the residual layer is detached from surface properties, and while some may argue it is not strictly part of the PBL, at the end of the day its inclusion depends on the definition used. It could also be useful to cite studies that have investigated the residual layer using lidar, for example Fochesatto et al., 2001.

   **The passage was revised to clarify the characteristics of the residual layer according to the reviewer's suggestions. The suggested Fochesatto citation has been added, as well as one that highlights the role of the residual layer in the morning growth phase, and another as an example of a group that defines the residual layer top as the PBLH. The new text can be found on lines 84-89. They read:**

   **"This residual layer is decoupled from the surface and lacks strong convection (though cases of weak mixing have been observed; Fochesatto et al., 2001), yet retains the history of mixing from the previous day. It persists above the stable boundary layer until both are entrained into the growing mixed layer the following day. Its presence and properties can strongly influence mixed layer development during the morning growth period (Blay-Carreras et al., 2014). Note that some researchers include the residual layer in their definition of the PBL, defining the PBLH as its top (e.g., Chu et al., 2022), in contrast to the definition used here."**

9. Line 87: The term "transition layer" is generally difficult (if not impossible) to capture by lidars due to overlap effects, which is also the case for MPD measurements.

**We interpreted this comment as raising two possible concerns: (1) overlap effects, which can limit measurements close to the surface, and (2) the distinction between the transition layer and the entrainment zone. In this paper, we define the transition layer as the potential temperature inversion and region of steep aerosol gradient at the top of the mixed layer, which can be observed with lidars under favorable conditions. In contrast, the entrainment zone is defined by negative buoyancy flux, which cannot be measured with the MPD or with lidars of similar temporal and vertical resolution. We do not consider the difficulty of measuring at low altitudes due to overlap effects to be directly pertinent here, but we discuss the MPD HSRL overlap in Section 3.2 (see response to comment #12). To make the distinction between the entrainment zone and the transition layer, we revised lines 96-100 to read:**

**"However, the buoyancy flux cannot be measured with the MPD or with lidars of comparable vertical and temporal resolution, and the entrainment zone spans a broader region than the sharp gradients typically observed by lidar (Brooks and Fowler, 2012). For this reason, this study avoids using the term "entrainment zone" in the context of lidar observations and instead uses "transition layer," which can be observed by lidar instruments in favorable conditions."**

10. Lines 91–98: The discussion here makes perfect sense, but could the authors provide references to support these statements?

**Citations were added to support the claims. In one location, the language was softened, changing "generally" to "often." The revised text now appears on lines 104–116. These lines read:**

**"The vertical extent of the mixed layer is called the MLH. The term PBLH refers more broadly to the full PBL depth, encompassing both the MLH in convective conditions and the stable boundary layer height in stable conditions (Stull, 1988). The mixing height is a related term primarily used for air quality forecasting, which refers to the height to which surface-based pollutants disperse, effectively representing the vertical extent of active turbulence (Seibert et al., 2000; Tucker et al., 2009). While it is often assumed to be equivalent to the MLH in convective conditions, turbulence does not always extend to the transition layer, leading to differences (Grimsdell and Angevine, 2002; Träumner et al., 2011; Schween et al., 2014). It is also defined in stable conditions when no mixed layer is present and usually differs from the stable boundary layer height (Tucker et al., 2009; Bonin et al., 2018). Despite the differences in definition, the same diagnostic methods are often appropriate for both the MLH and the mixing height in convective conditions. For example, wavelet transformations of the aerosol backscatter**

**profile have been used in studies retrieving both the MLH (Cohn and Angevine, 2000; Brooks, 2003) and the mixing height (Haeffelin et al., 2012; Schween et al., 2014). Many of the references (e.g., Seibert et al., 2000) discuss methods for diagnosing the mixing height. In this paper, the term MLH will be used for the lidar PBLH observations to distinguish them from the stable boundary layer height, typically below the minimum range of the lidars used in this study."**

11. Line 103: From a quick search, it seems that Tonopah is surrounded by several notable mountain ranges and peaks. How can it be described as a "flat basin" given its elevation of 1641 m? Additionally, there is a statement regarding Tonopah's MLH—has this been investigated in other studies or it is an empirical statement? What are the corresponding climate characteristics at the Colorado site? The measurement period seems short and may represent only limited atmospheric conditions. Finally, it would be helpful to clarify the typical aerosol types encountered at these sites (e.g., biomass, pollen, dust), especially because they are used as tracers for the MLH. Both Boulder and Tonopah are located near notable terrain features (e.g., Cheyenne Mountain for Boulder; Lone Mountain and Butler Siebert for Tonopah). Have the authors considered the impact of complex terrain on their MLH observations? Are there any previous studies investigating orography effects at these sites? Also, both sites are around 1600 m elevation, but their surface characteristics differ (forested vs. arid); could this influence the measurements or interpretation?

**The site descriptions have been revised to address the reviewer's concerns. We expanded the text to contain additional detail on topography, vegetation, aerosols, and climate for both locations. Two citations were added to support the statement regarding Tonopah's typical MLH values. The potential effects of the surrounding terrain are acknowledged, but they were not isolated in this study. To our knowledge, no previous studies have specifically investigated orographic effects at Tonopah or Boulder. The differences between the sites likely contribute to differences in MLH, but do not affect the lidar retrieval methodology. The elevation of the sites likewise does not affect the measurement or interpretation. The revised section is provided on lines 132-154.**

12. Line 115: HSRL measurements still require careful instrument calibration (e.g., molecular channel normalization, overlap correction near the surface) and quality control. It would also be helpful to mention the wavelength of operation.

**The text has been revised to clarify the distinction between the HSRL and elastic backscatter lidars and ceilometers. The MPD's HSRL wavelength (770 nm) was added, and the role of calibration and overlap correction was clarified. While most HSRLs require a differential overlap correction between the combined and molecular channels, the MPD measures this correction directly as part of its retrieval process (Stillwell et al., 2020; Hayman et al., 2024). Because the MPD combines DIAL and HSRL measurements, the $O_2$ online channel passes through**

the same receiver optics as the HSRL channels, allowing the calibration parameters to be measured continuously and accounted for within the retrieval itself (Hayman et al., 2024). The paragraph has also been reorganized for clarity. The passage, located on lines 159-169, reads:

"The HSRL technique measures vertical profiles of calibrated aerosol backscatter at a particular wavelength (770 nm for the MPD). In contrast, ceilometers and elastic backscatter lidars measure attenuated backscatter. An inversion technique (e.g., Fernald et al., 1972; Klett, 1981), which requires overlap correction, relies on assumptions about aerosol properties, and contains coupled backscatter and extinction terms, is required to isolate the aerosol backscatter, introducing further uncertainty. The HSRL relies only on an internal calibration of the spectral response of the two receiver channels and provides calibrated aerosol backscatter directly. Most HSRLs require a differential overlap correction between the combined and molecular channels, but the MPD measures this correction directly as part of its retrieval process (Stillwell et al., 2020; Hayman et al., 2024). The reduced reliance on assumptions and external calibrations makes the MPD HSRL more robust for high-quality MLH retrievals."

13. Line 116: How is the virtual potential temperature calculated? Are dry and moist air considered, and are clouds excluded? Is the water vapor mixing ratio used? How is pressure incorporated? This information, along with the relevant formulas, should be described in an appendix. In my opinion, since this is the first time the parcel method is applied to lidar data, a detailed methodological description is necessary.

An appendix (Appendix B) has been added that provides the complete derivation and description of the MPD-thermodynamic method. The appendix includes:
- Calculation of potential temperature from MPD temperature and pressure profiles.
- Conversion of absolute humidity, temperature, and pressure (measured by the MPD) to specific humidity.
- Conversion of potential temperature and specific humidity to virtual potential temperature.
- Description of how cloudy bins are masked.
- A complete methodological description of the implementation of the MPD-thermodynamic method.

The description of the MPD-thermodynamic method in the text was also shortened to avoid redundancy, as described in the response to the major comment. These additions provide the methodological detail and formulas necessary for applying the method to other lidar datasets.

14. Lines 118–119: When the authors say "the MPD had a minimum range of 318 m AGL and a vertical resolution of 150 m with 37.5 m range bin spacing," do they mean that the

MPD averages 4 bins to improve the signal-to-noise ratio (4 × 37.5 m = 150 m)? So, the meaningful profile resolution is 150 m, while the raw data points are spaced every 37.5 m and are noisier? Maybe I am missing something, please clarify.

**The passage has been revised to clarify the minimum range, vertical resolution, and range bin spacing. The revised text (lines 172-175) now reads:**

**"In the configuration deployed, the MPD had a minimum usable range of 318 m AGL (set by quality control; limited by detector recovery from the outgoing pulse; see Stillwell et al., 2025), an effective vertical resolution of 150 m (primarily set by the pulse length, though retrieval dependent; see Hayman et al., 2024), and a range bin spacing of 37.5 m from the 250 ns digitization, which oversamples the pulse. "**

15. Line 125: I suggest, if possible, to provide references for the Halo Streamline Pro and Vaisala instruments. Additionally, were any corrections applied for noise? What type of scan was performed (horizontal, vertical, or both)?

**A reference for the Halo Streamline DWL was added (Pearson et al., 2009) on line 185, which describes the predecessor design on which the Halo system is based. The instrument operated in vertical staring mode throughout the campaign, which is stated on line 189. References were not added for the radiosondes, weather station, or disdrometer, as these are best documented by manufacturer datasheets.**

**Regarding the noise of the DWL, this is addressed in Section 4.4, which describes the vertical velocity variance retrievals. Following Lenschow et al. (2000), the uncertainty in the vertical velocity variance was estimated, and periods where the uncertainty was comparable to or exceeded the variance were conservatively excluded from further analysis. This quality control step limited the dataset to unambiguous vertical velocity variance values. Three sentences (lines 290-292) were added to this section to make this clear. It reads:**

**"Uncertainty in the vertical velocity variance was estimated following Lenschow et al. (2000). Periods containing data within the mixed layer where the estimated uncertainty was comparable to or exceeded the vertical velocity variance were flagged and excluded from further analysis. To avoid bias, the approach erred on the side of discarding marginal data."**

16. Lines 130–131: Could the authors clarify why precipitation data are needed for the DWL? Does the instrument automatically stop or flag data during rain to avoid contamination, or is this used to filter MLH retrievals?

**Precipitation does not cause the instrument to stop working or trigger automatic data flags. Scattering from raindrops increases the measured vertical velocity variance in a way that makes it impossible to reliably identify the MLH. Because lidars cannot see well through precipitation, the MLH would not be observable even if precipitation periods were inferred directly from the DWL data. Since independent precipitation sensors were available at this site, we used those measurements to exclude affected periods from the analysis. The passage on lines 189-191 now reads:**

**"Precipitation data from this sensor and a disdrometer (OTT Parsivel$^2$) were used to identify periods of precipitation so they could be removed from the DWL retrievals, since scattering from raindrops increases the vertical velocity variance and prevents reliable MLH retrievals."**

17. Line 148: How are cloudy data removed from the lidar observations? Does the algorithm apply a threshold on the raw signal? Removing clouds is critical in lidar studies, and it would be helpful to clarify how the method distinguishes clouds from dense aerosol layers, which can also produce strong backscatter. Lines 163 and 186–187 suggest that some cloud-affected measurements remain in the dataset; please elaborate on how these are treated or flagged.

**We revised the text to clarify how cloudy data is treated. A citation to Colberg et al. (2022) has been added to line 208 to describe the cloud masking algorithm. Misidentification of dense aerosol layers as clouds is typically only a concern during heavy wildfire smoke events. Since these conditions did not occur during the field experiment, misidentification of clouds or dense aerosols is not expected to affect the results. The revised passage on line 208 now reads:**

**"The algorithm begins by detecting and removing cloudy data as described in Colberg et al. (2022)."**

**We also clarified how MLH estimates above low clouds were treated. MLH retrievals are excluded when the MLH estimate lies above clouds or precipitation, since retrievals above clouds are unreliable. MLH estimates below cloud bases were retained. Lines 229-230 now read:**

**"MLH estimates above low clouds or near precipitation were removed, since retrievals above these features are unreliable. MLH estimates below cloud bases were retained."**

**And line 257-258 now read:**

**"MLH estimates retrieved above clouds or precipitation were excluded as the temperature retrieval becomes unreliable in these conditions, while MLH estimates below cloud bases were retained."**

18. Appendix A – Is b the translation parameter? Please clarify, it may be confusing for the reader. I understand that rb and rt correspond to the bottom and top of the integration, but the role of b should be explicitly described. How is b defined within the piecewise function?

**Yes, b is the translation parameter, and that definition has been added. Short descriptions of the dilation, a, and translation, b, were also added. Also, the variables were changed so that the HWT is a function of height AGL, z, instead of range, r. The shift variable has been changed from z to u. Lines 639-640 now read:**

**"where z is the height AGL, a is the dilation (controlling the scale of the wavelet), b is the translation (shifting the wavelet along the height axis), and u is the shift variable for the convolution."**

19. Line 152: If you are referring to Dijkstra (1959), please cite it directly, even if it is already mentioned in de Bruine et al. (2017). This makes it easier for readers who encounter the algorithm for the first time. I think it is a pretty common algorithm in informatics, but not really in atmospheric physics. Also, could the authors clarify whether Pathfinder is a software package used for retrieving the MLH, or if the algorithm was implemented from scratch? Was it applied to 5-minute backscatter profiles? If yes (as mentioned in L53) was it attenuated backscatter? These details will help better understand the methodology.

**A direct citation to Dijkstra (1959) has been added. We also clarified that Pathfinder refers to the method described by de Bruine et al. (2017), not a software package. The Pathfinder method was implemented from scratch for this study, following the description in de Bruine et al. (2017), and the code has been provided with this manuscript. The algorithm was applied to the 5-minute average calibrated aerosol backscatter profiles from the MPD, rather than attenuated backscatter as in the original study. We have also added a line referring the reader to the attached code for the details of the implementation. Lines 212-214 now read:**

**"After the HWT, Dijkstra's algorithm (Dijkstra, 1959) is applied to track layers over time using the Pathfinder method described by de Bruine et al. (2017)."**

**And lines 216-218 now read:**

**"For this study, the Pathfinder method was independently implemented and applied to the HWT of calibrated aerosol backscatter profiles averaged over five**

**minutes. Implementation details are provided in the accompanying code (see Code Availability section)."**

20. Lines 156–159: The range for the top limiter seems quite large. Is it empirically set? Also, how does the algorithm handle cases where aerosol layers exist close to the top of the mixing layer, or when strong horizontal winds enhance turbulence and lead to unusually high MLH values?

**The range of the top limiter is large, but it is only a search bound rather than a constraint on the retrieved MLH. The top limiter is set to the HRRR MLH + 500 m to reduce the spurious detection of lofted residual layers, and after midday, it is set to 6 km, which exceeds the deepest MLH observed in the dataset and effectively removes the constraint. This allows deep MLH values in the afternoon. In principle, an unusually deep morning MLH could fall outside this bound if the HRRR MLH estimates were biased low, but we did not encounter such cases in this study. The limiter functions as a flexible and adaptive reference rather than a rigid threshold, which reduces the number of empirical constraints that it requires, improving the algorithm's ability to generalize. A sentence has been rewritten to clarify on lines 223-226, and it reads:**

**"This limiter only bounds the search region. It helps to avoid early morning selection of lofted residual layers instead of a shallow mixed layer and is effectively disabled after midday since 6 km exceeds the deepest MLH observed in the dataset. The limiter guides the search toward the correct MLH by using the HRRR MLH as a flexible, adaptive reference rather than imposing a rigid cutoff."**

21. Line 163: Again, it is unclear how low clouds are identified and removed from the lidar data. Are specific thresholds applied to the signal? How are these distinguished from dense aerosol layers that could produce similarly strong backscatter? A clear description is needed.

**This concern was also raised in comment #17. We have clarified how cloudy data are treated. The revision on line 208 describes how the clouds are detected and masked from the dataset, and the revision on lines 229-230 describes how MLH estimates are handled in the presence of clouds. As far as the distinction between clouds and dense aerosols, this is usually only an issue during dense wildfire smoke, which was not seen during this dataset.**

22. Line 176: Please clarify whether this weather station is the same as the one mentioned in line 130. If it is integrated into the MPD system, this should be also stated in Section 3.3.

**The built-in weather station is not the same as the one on line 130. It is the built-in station that is a part of every MPD instrument (as described in Spuler et al., 2021),**

**and we consider it a part of the MPD system. To avoid confusion, we added a short clarification sentence to Section 3.2. The sentence, on lines 179-180, reads:**

**"The MPD also includes a built-in weather station (Lufft WS300), which provides the surface observations required for the MPD-thermodynamic method."**

23. Lines 178–187: It seems that many of these thresholds and limits were set empirically. A visual representation (e.g., a diagram or flowchart) would help the reader understand the workflow. This is especially useful for someone applying the thermodynamical parcel method to their own lidar data. For instance, a figure showing how the virtual potential temperature is calculated, including how different time resolutions for humidity and temperature are handled, would make the method more transparent. I would suggest, if not included in the main text, consider adding it to the appendix.

**A visual representation (Figure B1) and a complete methodological description were added to Appendix B, along with guidance for applying the approach to other datasets from remote sensing instruments. Figure B1 shows how the offset and top limiter thresholds work in practice, showing the MPD-thermodynamic workflow.**

**The MPD produces data at 5 minute intervals, so there is no mismatch in the temporal resolution. The effective temporal resolution is different for each data product, and it depends on the Poisson Total Variation retrieval (Hayman et al. 2024). Poisson Total Variation adapts to the signal-to-noise ratio, so the effective temporal resolution changes based on the signal, leading to longer temporal resolution for noisier retrievals, like temperature. This was clarified on lines 176-179, which now read:**

**"Aerosol backscatter, water vapor, and temperature profiles were produced at five-minute intervals, with effective temporal resolutions of 5 minutes for aerosol backscatter, 10 minutes for water vapor, and 40 minutes for temperature, as determined by the retrieval processing described in Hayman et al. (2024)."**

**The thresholds used in the method are now explicitly described as empirical both in the main text (line 251) and in Appendix B (lines 684, 687, 698).**

24. Line 193: The parcel method using a 1 K offset from the surface value, is applied in the same way as for the MPD? So that the initial condition or reference for both methods is consistent (or, if different, how it differs).

**Yes, the parcel method using a 1 K offset from the surface value was applied in the same way as for the MPD. To remove any ambiguity and avoid redundancy, the sentence has been removed, and the information has been added. A detail clarifying that surface measurements were taken from the first upper-air**

**observation by the sonde has also been added. This passage, on lines 263-266, now reads:**

**"This study used two methods: the bulk Richardson method (Seidel et al., 2012; Guo et al., 2021) and the parcel method with a 1 K offset, applied identically to the MPD-thermodynamic method, except the surface virtual potential temperature was taken from the first valid in-air sonde level (typically about 10 m AGL), since surface observations at launch are often unreliable. "**

25. Lines 194–201: The comparison between radiosonde–Richardson and MPD–aerosol MLH retrievals is reasonable for daytime convective conditions, but it can diverge in transition periods or multi-layer situations. After sunset or in late morning, elevated well-mixed layers or residual aerosol layers can persist aloft, so the Richardson method (sensitive to turbulence) may detect a different top than the aerosol-based MLH. The authors partly mention this (L202–203), but it should be emphasized that their comparison is valid for daytime convective periods, as the algorithm does not run at night. It would also help to specify until what local time the algorithm provides retrievals. Is it 20:00 local time? Additionally, in my opinion it is worth expanding on the Richardson method in the appendix, similar to the treatment for Wavelet Haar and the wind lidar, to clarify its application.

**We have clarified the scope and limitations of the MPD retrievals and expanded the description of the bulk Richardson method as follows:**

**Because the MPD-aerosol method assumes aerosols are passive tracers of the MLH, the assumptions of this retrieval method are not valid at night. For this reason, the MPD-aerosol method was only applied between local sunrise and local sunset. This is now stated on lines 232-233:**

**"Accordingly, this method is only applied between local sunrise and local sunset."**

**Additionally, in Section 5.2, a sentence was added to clarify that the 10:00 and 15:00 local time radiosonde launches correspond to daytime convective conditions, within the valid operating times of the retrieval algorithms. The sentence, on lines 423-425, now reads:**

**"Both launch times occurred during daytime convective conditions within the valid operating range of the MPD retrieval algorithms (sunrise to sunset)."**

**To be consistent across methods, the time series comparisons in Section 5.3 were restricted to daytime conditions. A clarifying sentence was added to lines 506-507 that reads:**

**"Because the MPD-aerosol method only operates between sunrise and sunset, all comparisons were limited to this daytime period."**

**Appendix C was also added to present the math for the bulk Richardson number method to mirror the treatments of the HWT and the DWL methods. It contains:**
- **Conversion of the temperature, pressure, relative humidity, and wind speeds from the radiosondes to the bulk Richardson number**
- **A discussion of the physical meaning of the bulk Richardson number**
- **The use of the first valid in-air data from the sonde for surface temperature and humidity, and the use of a surface weather station for wind speeds.**
- **An explanation of the critical threshold (0.25), following Seibert et al. (2000), Seidel et al. (2012), and Guo et al. (2021).**

**Additionally, Section 4.3 was rewritten to remove redundancy, as many of the details were moved to the appendix. This section, on lines 269-276, now reads:**

**"Details of the bulk Richardson number calculation, including the formulation of virtual potential temperature and the critical thresholds, are provided in Appendix C. In brief, the MLH is diagnosed as the lowest altitude where the bulk Richardson number exceeds a critical value of 0.25 (Seibert et al., 2000; Seidel et al., 2012; Guo et al., 2021)."**

**Together, these revisions clarify the valid operating range of the retrievals and provide the expanded methodological detail requested for the Richardson method.**

26. Lines 220–221: It should be briefly clarified how virga was manually removed from the lidar data. For example, was a threshold applied to the attenuated backscatter or signal strength? Providing this detail helps readers understand how aerosol layers were distinguished from falling precipitation or virga.

**Virga was identified by manual inspection of the vertical velocity and backscatter fields, where it appears as a region of consistent negative vertical velocity and enhanced backscatter underneath a cloud. Automated virga detection thresholds were avoided because they risk excluding strong downdrafts. The affected time periods were flagged and excluded from further analysis. This clarification was added on lines 299-301, which now read:**

**"Virga was identified through inspection of the vertical velocity and backscatter fields, where it appears as a region of consistent negative vertical velocity and enhanced backscatter beneath a cloud. The affected periods were excluded from the analysis."**

27. Lines 228–229: Could you clarify whether the parcel method was also applied to the HRRR virtual potential temperature fields? If so, this strengthens the need to include the full description and formulas of the parcel method in the appendix.

**Yes, the parcel method was also applied to the HRRR virtual potential temperature fields, identically to the MPD-thermodynamic method. A sentence was modified to include this explicit statement in Section 4.5 and to reference Appendix B. The revised sentence, on lines 310-312, reads:**

**"These profiles, together with the HRRR surface fields, were used with the 1 K offset parcel method (applied identically to the MPD-thermodynamic method; see Appendix B), enabling direct comparisons."**

28. The authors have done a great job discussing the 6 and 8 September case studies. The manuscript clearly contrasts different boundary layer conditions and aerosol structures, effectively demonstrating how the methodology performs.

**We thank the reviewer for this positive feedback.**

29. Section 5.2 is particularly valuable, as it compares the parcel method applied to different instruments—radiosonde (in-situ) and lidar (remote sensing). These instruments measure different tracers for MLH, making the comparison insightful, and the authors describe the results well. However, clarification on collocation is needed: Were the lidar products time-averaged around the radiosonde launch? For example, was the closest 5-min backscatter profile used for the MPD-aerosol method, 10-min water vapor, and 40-min temperature for the MPD thermodynamic method?

**We clarified the temporal resolution in Section 3.2 and the collocation in Section 5.2.**

**The clarification of the temporal resolution is on lines 176-179 and reads:**

**"Aerosol backscatter, water vapor, and temperature profiles were produced at five-minute intervals, with effective temporal resolutions of 5 minutes for aerosol backscatter, 10 minutes for water vapor, and 40 minutes for temperature, as determined by the retrieval processing described in Hayman et al. (2024)."**

**The clarification for the collocation is on lines 425-427 and reads:**

**"For each radiosonde, the lidar profile and MLH estimate used in the comparison were those closest in time to when the radiosonde ascent passed through the lidar-derived MLH, instead of the launch time, which could introduce a mismatch due to the radiosonde ascent time."**

30. L355: Do you mean that the MLH retrieved from the parcel method is on average 250 m higher than the MLH from the Richardson method? If so, why this difference occurs? Both methods detect turbulent layering, but the Richardson method responds directly to shear and turbulence, whereas the parcel method may follow the thermodynamic profile, potentially resulting in systematically higher MLH estimates.

**We thank the reviewer for the suggestion. While both methods are linked to thermodynamic layering, the bulk Richardson method also accounts for turbulence generation by wind shear. In practice, the systematic difference arises primarily from the 1 K offset applied to the parcel method. An explanation for the 250 m difference has been added on lines 445-450:**

**"This difference arises primarily from the 1 K offset applied to the parcel method. The standard parcel method (no offset) corresponds to the height where the bulk Richardson number is 0 (see Appendix C). With the 1 K offset applied, the parcel method tends to diagnose an MLH about 150–200 m higher on average. The bulk Richardson method only produces higher MLH values in cases of very high wind speed. During morning growth, the offset parcel method is also more likely to select the top of a residual layer, producing larger outliers (300–1000 m) that raise the mean difference to about 250 m when considering all radiosonde launches."**

31. L346-349: The description of the different comparisons in each column could be clarified. From the figure, it seems that (a) and (d) show all conditions, (b) and (e) show clear-sky conditions, and (c) and (f) show clear-sky conditions at 15:00 local time. I believe it would help the reader if this were explicitly stated in the text. Also, please clarify how clouds are defined or detected in your MPD retrievals, since this affects which points are included in the clear-sky subsets.

**The passage describing the figure was revised to clarify what each panel shows and to define the clear-sky designation, as suggested. The modified passage is on lines 432-438.**

**"Figure 4 shows scatter plots comparing MPD and radiosonde MLH retrievals across increasingly restrictive subsets. Panels (a)–(c) compare the MPD-aerosol to the bulk Richardson method, while panels (d)–(f) compare the MPD-thermodynamic to the radiosonde offset parcel method. Panels (a) and (d) show all comparisons, panels (b) and (e) show clear-sky conditions, and panels (c) and (f) show clear-sky conditions at 15:00 local time. Clear-sky cases may include sparse cloud cover, provided none are directly above the MPD at launch, as neither MPD method can retrieve the MLH when clouds are present at or below it.Clear-sky conditions are defined as those with no cloud detected directly above the lidar during the time the radiosonde ascent passed through the MPD-derived MLH, though nearby clouds outside the lidar beam may have been present."**

**Further details on cloud detection are provided in the response to comment #17.**

32. Fig 5: Consider maybe adding a small legend for the two lines at the bottom right to make it easier for the reader to distinguish them quickly. Also, using the same solid black line for both radiosonde retrievals (panels a, c) and MPD retrievals (panel b) can be confusing, even if explained in the caption.

**Per the reviewer's suggestion, the legend has been added to panels (a) and (c), and the MPD aerosol backscatter profile in panel (b) was changed to a blue dashed line. The caption was updated accordingly.**

**"Figure 5. Radiosonde and MPD profiles for the radiosonde launched at 22:15 UTC on 26 July 2023. The figure shows (a) virtual potential temperature from the radiosonde and MPD, (b) aerosol backscatter coefficient from the MPD, (c) relative humidity from the radiosonde and MPD, and (d) bulk Richardson number from the radiosonde. Horizontal dotted lines indicate MLH retrievals: 1—Radiosonde offset parcel method, 2—Radiosonde bulk Richardson method, 3—MPD-thermodynamic method, and 4—MPD-aerosol method. The solid black lines represent radiosonde profiles in panels (a), (c), and (d), and the dashed blue lines represent MPD profiles in panels (a), (b), and (c). All heights are AGL."**

33. Section 5.3: It seems that each point in Figure 6 represents roughly 2 hours, given that 0 corresponds to 06:00 LT and 1 to 20:00 LT (14 hours divided by 9 points). Is it true? It would be helpful to clarify the model's time step/resolution, and this information, along with the spatial resolution, should be included in Section 4.5.

**Each point in Figure 6 represents one-tenth of a normalized day, which corresponds to about 75-85 minutes depending on the day length. We have added a sentence (lines 504-505) to clarify this. We have also clarified the temporal, spatial, and vertical resolutions of the HRRR data. The horizontal resolution and the temporal resolution were already listed as 3 km and 1 hour (lines 303-304). Lines 307-312 now clarify that the native PBLH estimate is a continuous variable in meters AGL and that the vertical resolution of the virtual potential temperature profiles is 25 hPa (~250 m).**

**Line 307-312 now read:**

**"The native PBLH estimate is reported as a continuous variable in meters AGL. Additionally, a second MLH estimate was computed from the HRRR virtual potential temperature profiles, which, in the format used here, are available every 25 hPa in the boundary layer (corresponding to about 250 m vertical spacing at an elevation of 1600 m above sea level, similar to the Tonopah and Boulder sites). These profiles, together with the HRRR surface fields, were used with the 1 K**

offset parcel method (applied identically to the MPD-thermodynamic method; see Appendix B), enabling direct comparisons."

**Lines 504-505 now read:**

**"This corresponds to a spacing of about 75–85 minutes, depending on day length."**

Reviewer #2

1. How many days were finally used for the time series comparison? Please indicate a number.

   **There were 52 days in the dataset. Of these, 2 were affected by Hurricane Hilary, leaving 50 days taken for the MPD and HRRR method comparisons. Excluding an additional 17 days that had signal dropout in the DWL data leaves 33 days. This clarification has been added on lines 507-508, which now read:**

   **"After excluding the two days affected by Hurricane Hilary, the time series comparison included 50 days for the MPD and HRRR methods. For DWL comparisons, an additional 17 days with signal dropouts were excluded, leaving 33 days."**

2. Will the MPD lidar be commercially available soon for the community?

   **The commercialization of the MPDs is outside the scope of this paper, so no changes were made to the manuscript. An update on the MPD can be obtained by contacting the UCAR Technology Transfer Office at ipinfo@ucar.edu.**

Additional Changes

In addition to the changes in response to reviewer comments, the following changes were made.

1.  **A citation (Christopoulos et al., 2025) was added to line 47 to acknowledge related work that strengthens the context for this study.**

2.  **The following sentence was added to the discussion (lines 574-576) to highlight potential applications for the MPD:**

    **"In addition to the continuous improvements to hardware and processing, other potential advancements to the MPD data products include the retrieval of variables relevant to convection (e.g., Hoffman and Demoz, 2025), such as equivalent virtual potential temperature, CAPE (convectively available potential energy), and CIN (convective inhibition)."**

3.  **The color scale of the upper plot in Figure 7 was changed to match the color scales of the upper plot in Figure 2 and Figure 3.**

4.  **In Appendix A, the equations were rewritten as functions of height AGL, $z$, instead of range, $r$, to match the notation used in the other appendices. The shift variable in the convolution was changed from $z$ to $u$.**

5.  **On line 381, the phrase "a warm, aerosol-dense plume" was changed to "an aerosol-dense plume."**